# Accurate KV Cache Eviction via Anchor Direction Projection for Efficient LLM Inference

**Zijie Geng[1], Jie Wang[1]\*, Ziqi Liu[1], Feng Ju[1], Yiming Li[2], Xing Li[2], Mingxuan Yuan[2],**
**Jianye Hao[2,3], Defu Lian[1], Enhong Chen[1], Yongdong Zhang[1], Feng Wu[1]**
[1]University of Science and Technology of China
[2]Noah's Ark Lab, Huawei          [3]Tianjin University
{zijiegeng,ziqiliu,fengju}@mail.ustc.edu.cn, jiewangx@ustc.edu.cn

## Abstract

Key-Value (KV) cache eviction—which retains the KV pairs of the most important tokens while discarding less important ones—is a critical technique for optimizing both memory usage and inference latency in large language models (LLMs). However, existing approaches often rely on simple heuristics—such as attention weights—to measure token importance, overlooking the spatial relationships between token value states in the vector space. This often leads to suboptimal token selections and thus performance degradation. To tackle this problem, we propose a novel method, namely **AnDPro** (**An**chor **D**irection **Pro**jection), which introduces a projection-based scoring function to more accurately measure token importance. Specifically, AnDPro operates in the space of value vectors and leverages the projections of these vectors onto an *"Anchor Direction"*—the direction of the pre-eviction output—to measure token importance and guide more accurate token selection. Experiments on 16 datasets from the LongBench benchmark demonstrate that AnDPro can maintain $96.07\%$ of the full cache accuracy using only $3.44\%$ KV cache budget, reducing KV cache budget size by $46.0\%$ without compromising quality compared to previous state-of-the-arts.

## 1 Introduction

Large language models (LLMs) have revolutionized human society by reshaping the way we process information [1–5]. Modern LLMs are primarily built upon the Transformer architecture, with self-attention as their core mechanism [6]. However, the self-attention mechanism inherently involves quadratic computational complexity, which grows dramatically as the sequence length increases [7]. To address this bottleneck, the Key-Value (KV) cache mechanism has been introduced to store and reuse intermediate results—i.e., the KV pairs in the self-attention mechanism—to reduce the computational complexity from quadratic to linear [8]. As the demand for processing longer sequences grows, however, the KV cache size expands proportionally, leading to substantial increases in both GPU memory consumption and I/O latency [9]. Therefore, effective KV cache management is critical for practical LLM scaling and deployment.

Recent studies have revealed that LLM self-attention exhibits inherent sparsity, with only a small subset of tokens making a substantial contribution to the model's output [10–12]. This insight has sparked the development of a series of KV cache eviction strategies, which identify and retain the KV pairs of the most important tokens while discarding less important ones [13–15]. For instance, H2O [9] greedily eliminates KV pairs based on their cumulative attention weights during the generation phase. SnapKV [16] employs an observation window to generate queries and uses the obtained attention weights to measure the importance of prompt tokens during the prefilling phase. Ada-KV [17] further

---

*Corresponding author.

39th Conference on Neural Information Processing Systems (NeurIPS 2025).

refines this paradigm by jointly identifying important tokens across attention heads within each Transformer layer. By selectively removing less critical tokens, these approaches can effectively optimize both memory and time efficiency without significantly compromising the performance.

The aforementioned approaches essentially rely on an importance criterion—which we refer to as a "scoring function"—to determine token retention priority. Most of them use attention weights as the scoring function, assuming that tokens with higher attention weights should be retained preferentially. Although this assumption is intuitive, they overlook the spatial relationships between token value states in the vector space, which play a critical role in how these states interact to influence the model's output. This often leads to suboptimal token selection decisions and thus performance degradation. Therefore, effective KV cache management requires advancing beyond attention-only heuristics by developing scoring functions that explicitly account for the vector space geometry.

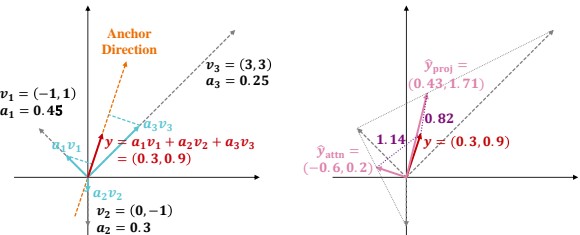

(a) Anchor direction projection     (b) Post-eviction output

Figure 1: **A diagram of the projection-based KV cache eviction.** (a) Consider three tokens with **value states** $v_1$, $v_2$, $v_3$, and **attention weights** $a_1$, $a_2$, $a_3$. The **pre-eviction output** is denoted as $y$. Our approach computes **the projections of** $a_i v_i$ onto the **"anchor direction"** (i.e., the direction of $y$). (b) With a budget of 2, the attention-based eviction approach selects $v_1$ and $v_2$, resulting in a **post-eviction output** $\hat{y}_{attn}$ and an error of $\|y - \hat{y}_{attn}\|_2 = \mathbf{1.14}$. Our projection-based approach selects $v_1$ and $v_3$, resulting in a **post-eviction output** $\hat{y}_{proj}$, with a smaller error of $\|y - \hat{y}_{proj}\|_2 = \mathbf{0.82}$.

To tackle this problem, we propose a novel method, namely **AnDPro** (**An**chor **D**irection **Pro**jection), which introduces a projection-based scoring function to more accurately measure token importance. Specifically, we formulate KV cache eviction as a combinatorial optimization problem and then transform it into a sparse optimization formulation through mild relaxations. The theoretical and intuitive analysis of this problem motivates us to introduce an *"anchor direction"* in the vector space, i.e., the direction of the pre-eviction output. Intuitively, this anchor direction guides the retention of the most important semantic information, leading us to use the projections of token value states onto this direction to measure token importance (see Figure 1). Experiments on 16 datasets from the LongBench benchmark demonstrate that AnDPro maintains 96.07% of the full cache accuracy using only 3.44% KV cache budget, reducing KV cache size by 46.0% without compromising model quality compared to previous state-of-the-arts.

## 2 Related Work

LLMs have been widely applied across various domains and tasks [18, 19]. Among the numerous research directions related to LLMs, our work is most closely aligned with LLM inference acceleration [20–25], particularly in the areas of KV cache management, eviction, and budget allocation.

**KV Cache Management** plays a crucial role in optimizing both memory and time efficiency in LLM inference. In a recent survey by Li et al. [13], KV cache management techniques are categorized into three levels: token-level, model-level, and system-level strategies. Our work is primarily related to token-level KV cache optimization, specifically KV cache eviction and KV cache budget allocation. Other token-level methods include KV cache merging [26–30], KV cache quantization [31–37], and KV cache low-rank decomposition [38–41].

**KV Cache Eviction** approaches selectively retain the most important KV pairs and discard the less critical ones. Some methods, such as StreamingLLM [14], FastGen [11], and recent studies [20, 21], are designed based on certain patterns and fixed rules, such as the sliding windows, for KV cache eviction. Other approaches, such as H2O [9], Scissorhands [10], and recent studies [42, 43, 22, 44], leverage attention scores as a criterion for selecting the most important KV pairs from the cache. SnapKV [16] introduces an observation window to generate queries, using accumulated attention scores to evaluate token importance in prefilling. Ada-KV [17] refines this framework by adaptively allocating the cache budget across attention heads based on the attention scores. More recently,

CriticalKV [45] combines attention scores with the norm of value vectors as a better scoring function. Our work builds upon this line of research, and draws inspiration from SnapKV and Ada-KV.

**KV Cache Budget Allocation** focuses on allocating the budget across different components of the model, such as layers and attention heads. Layer-wise allocation techniques include PyramidInfer [46] and DynamicKV [47]. Head-wise budget allocation methods include Ada-KV [17], HeadKV [48], and DuoAttention [49]. These methods are orthogonal to our work, as we focus on retaining important KV pairs for any layer- or head-level budget allocation. In this work, we implement our approach to achieve an adaptive budget allocation across attention heads.

# 3 Our Approach

In this section, we introduce our proposed method, **AnDPro** (**An**chor **D**irection **Pro**jection). Section 3.1 begins by formalizing the KV cache eviction problem to provide a theoretical foundation for deeper analysis. In Section 3.2, we present a theoretical exploration of the optimization problem, which motivates the general form of projection-based scoring function. Section 3.3 further derives a practical form of the scoring function through an intuitive analysis. Finally, in Section 3.4, we provide the implementation details of AnDPro. All the proofs can be found in Appendix A.

## 3.1 Problem Formulation

This section aims to answer the following question:

**Q1:** *How to formulate the KV cache eviction problem in a manner that enables in-depth analysis on the design of an appropriate scoring function for KV cache eviction?*

Our objective is to select a subset of tokens from the input prompt to retain, discarding the others, while approximating the original outputs as accurately as possible. In this section we consider one single attention head. Consider an input sequence $X = [x_1, \cdots, x_n]$, where $n$ represents the context length, i.e., the number of tokens in the prompt. [1] Let $\boldsymbol{x}_i \in \mathbb{R}^d$ denote the embedding vector of the $i^{\text{th}}$ token, where $d$ denotes the embedding dimension. Let $\mathbf{W}^Q, \mathbf{W}^K, \mathbf{W}^V \in \mathbb{R}^{d \times d}$ be the parameter matrices for the query, key, and value projections, respectively. For the $i^{\text{th}}$ token, its query $\boldsymbol{q}_i$, key $\boldsymbol{k}_i$, and value states $\boldsymbol{v}_i$ are given by:

$$\boldsymbol{q}_i = \mathbf{W}^Q \boldsymbol{x}_i, \quad \boldsymbol{k}_i = \mathbf{W}^K \boldsymbol{x}_i, \quad \boldsymbol{v}_i = \mathbf{W}^V \boldsymbol{x}_i. \tag{1}$$

Let $\boldsymbol{q}$ be a query given from an observation window, which will be further detailed in Section 3.4. The pre-eviction output of the attention head is given by:

$$\boldsymbol{y} = \sum_{i=1}^{n} a_i \boldsymbol{v}_i, \quad \text{where } a_i = \underset{1 \leq i \leq n}{\text{Softmax}}(\boldsymbol{q}^\top \boldsymbol{k}_i). \tag{2}$$

Let $S \subset [n]$ be the set of tokens selected to retain. The post-eviction output, i.e., the output derived when only retaining KV pairs corresponding to tokens $i \in S$, is then given by:

$$\hat{\boldsymbol{y}} = \sum_{i \in S} \hat{a}_i \boldsymbol{v}_i, \quad \text{where } \hat{a}_i = \underset{i \in S}{\text{Softmax}}(\boldsymbol{q}^\top \boldsymbol{k}_i). \tag{3}$$

Our objective is that the post-eviction output approximates the pre-eviction output as accurately as possible. It is intuitive to use the squared error as a metric to quantify the eviction loss:

$$\mathcal{L}(\boldsymbol{y}, \hat{\boldsymbol{y}}) \triangleq \frac{1}{2} \|\boldsymbol{y} - \hat{\boldsymbol{y}}\|_2^2. \tag{4}$$

Additionally, we impose a sparsity constraint $|S| \leq k$ on the number of retained tokens. The following proposition provides a more convenient expression of the post-eviction output to aid the further analysis.

**Proposition 1.** *Let $S \subset [n]$ be the set of retained tokens, and define the indicator function*

$$z_i \triangleq \begin{cases} 1, & i \in S, \\ 0, & i \notin S \end{cases} \tag{5}$$

---

[1] Throughout this paper, all vectors are presented as column vectors. We use the index $i$ and its corresponding token $x_i$ interchangeably, given that no ambiguity arises.

*to represent whether the $i^{th}$ token is retained. Then, the post-eviction output can be expressed as:*

$$\hat{\boldsymbol{y}} = \frac{\sum_{i=1}^{n} z_i a_i \boldsymbol{v}_i}{\sum_{i=1}^{n} z_i a_i}. \tag{6}$$

Based on Proposition 1, we can formulate the KV cache eviction problem as follows:

$$\min_{\boldsymbol{z} \in \{0,1\}^n} \frac{1}{2} \left\| \boldsymbol{y} - \frac{\sum_{i=1}^{n} z_i a_i \boldsymbol{v}_i}{\sum_{i=1}^{n} z_i a_i} \right\|_2^2, \quad \text{s.t.} \quad \sum_{i=1}^{n} z_i \leq k. \tag{P0-C}$$

In the realm of sparse optimization, a prevalent approach for dealing with the sparsity constraint is to introduce an $l_1$-norm regularization [50]. When dealing with binary variables, the $l_1$-norm is equivalent to $l_0$ norm, which counts the number of non-zero elements. The regularized form of Problem (P0-C) is then expressed as:

$$\min_{\boldsymbol{z} \in \{0,1\}^n} \frac{1}{2} \left\| \boldsymbol{y} - \frac{\sum_{i=1}^{n} z_i a_i \boldsymbol{v}_i}{\sum_{i=1}^{n} z_i a_i} \right\|_2^2 + \lambda \sum_{i=1}^{n} z_i. \tag{P0-R}$$

Although Problem (P0-R) is in a concise form, it remains $\mathcal{NP}$-hard, making exact solution computationally infeasible. Given the context of LLM inference, where a solution needs to be found within milliseconds, we seek to identify high-quality solutions quickly and approximate the problem's optimal solution. To this end, we explore mild relaxations of (P0-R) to derive a scoring function that can facilitate fast filtering for efficient KV cache eviction. Notice that directly relaxing the binary variables $z_i$ in (P0-R) to continuous values would lead to problem degeneration and results in trivial solutions. Specifically, when $z_i$ are relaxed, they can be scaled arbitrarily without affecting the first term in the objective function. If the values of $z_i$ are scaled sufficiently small, the second term in the objective function will diminish to zero, yielding trivial solutions.

We introduce new variables $s$ and $\beta_i$, defined by

$$s \triangleq \sum_{i=1}^{n} z_i a_i \quad \text{and} \quad \beta_i \triangleq \frac{z_i}{s}. \tag{7}$$

Then, we obtain:

$$\frac{\sum_{i=1}^{n} z_i a_i \boldsymbol{v}_i}{\sum_{i=1}^{n} z_i a_i} = \frac{\sum_{i=1}^{n} \beta_i a_i \boldsymbol{v}_i}{\sum_{i=1}^{n} \beta_i a_i} \quad \text{and} \quad \sum_{i=1}^{n} \beta_i a_i = 1. \tag{8}$$

Consequently, Problem (P0-R) can be reformulated as:

$$\min_{\boldsymbol{z} \in \{0,1\}^n} \frac{1}{2} \left\| \boldsymbol{y} - \sum_{i=1}^{n} \beta_i a_i \boldsymbol{v}_i \right\|_2^2 + \lambda s \sum_{i=1}^{n} \beta_i,$$
$$\text{s.t.} \quad \sum_{i=1}^{n} \beta_i a_i = 1, \qquad s\beta_i = z_i, \quad i = 1, \cdots, n. \tag{9}$$

In this reformulation, we can select an arbitrarily small $s$, which would lead to extremely small $z_i$. As a result, the regularization term would approach zero, and smaller values of $s$ would always yield better objective function values. Moreover, if we impose a lower bound on $s$, the optimal solutions will always set $s$ take to this lower bound. Therefore, we fix $s$ as a constant satisfying $0 < s \leq 1$, and perform a convex relaxation of the binary variables $z_i$. This leads to the following optimization problem:

$$\min_{\boldsymbol{\beta} \in [0, \frac{1}{s}]^n} \frac{1}{2} \left\| \boldsymbol{y} - \sum_{i=1}^{n} \beta_i a_i \boldsymbol{v}_i \right\|_2^2 + \lambda s \sum_{i=1}^{n} \beta_i, \quad \text{s.t.} \quad \sum_{i=1}^{n} \beta_i a_i = 1. \tag{P1}$$

**A1:** *Through mild relaxations, we derive Problem (P1), based on which we further work on the design of the scoring function in the subsequent sections.*

**Remark 1.** *We would like to re-emphasize that our goal is not to achieve an exact solution to the original problem, which is complex enough so that an exact solution is intractable. Instead, we aim to leverage this analysis to provide insightful motivations for the design of scoring functions.*

## 3.2 Projection-Based Scoring Function: General Form

Although Problem (P1) offers a concise formulation, solving it directly during the LLM inference process remains infeasible. Inspired by the "screening rules" in sparse optimization [51], we now investigate the design of a closed-form scoring function that can identify the active variables in Problem (P1), i.e., those with non-zero coefficients in the optimal solution. In this way, we can retain the active variables without explicitly solving the entire optimization problem. Thus, this section aims to answer the following question:

**Q2:** *Is it possible to find a closed-form scoring function that identifies the active variables for Problem (P1)? If so, what is the ideal mathematical form of such a scoring function?*

Let $\boldsymbol{\beta}^*$ denote an optimal solution to Problem (P1). In this section, we will show that under certain conditions, it is possible to determine that $\beta_i^* = 0$ for some $i \in [n]$, implying that the corresponding tokens can be removed from the cache. To further investigate the problem, we introduce a new set of variables, $\boldsymbol{u} \triangleq \boldsymbol{y} - \sum_{i=1}^n \beta_i a_i \boldsymbol{v}_i$. We can then reformulate Problem (P1) as:

$$\min_{\boldsymbol{\beta} \in [0, \frac{1}{s}]^n} \quad \frac{1}{2} \|\boldsymbol{u}\|_2^2 + \lambda s \sum_{i=1}^n \beta_i, \tag{P-Primal}$$

$$\text{s.t.} \quad \boldsymbol{u} = \boldsymbol{y} - \sum_{i=1}^n \beta_i a_i \boldsymbol{v}_i, \quad \sum_{i=1}^n \beta_i a_i = 1$$

We can now derive the dual problem of (P-Primal), as presented in the following theorem.

**Theorem 1.** *The dual problem to Problem (P-Primal) is given by:*

$$\sup_{\boldsymbol{\eta}, \zeta} \quad -\frac{1}{2} \|\boldsymbol{\eta}\|^2 + \boldsymbol{\eta}^\top \boldsymbol{y} - \zeta + \frac{1}{s} \sum_{i=1}^n \rho_i(\boldsymbol{\eta}, \zeta), \tag{P-Dual}$$

*where $\boldsymbol{\eta}^* \in \mathbb{R}^n$ and $\zeta \in \mathbb{R}$ are dual variables, and*

$$\rho_i(\boldsymbol{\eta}, \zeta) \triangleq \min\left(0, \lambda s - a_i\left(\boldsymbol{\eta}^\top \boldsymbol{v}_i - \zeta\right)\right). \tag{10}$$

Problem (P-Primal) is convex, and its constraints are affine. According to Slater's condition [52], as long as (P-Primal) is feasible, strong duality holds. The reason why we care about (P-Dual) lies in the utilization of the optimal conditions, which can provide valuable insights into the optimal solution $\boldsymbol{\beta}^*$. Specifically, by applying the Karush-Kuhn-Tucker (KKT) optimal conditions, we can derive the following theorem:

**Theorem 2.** *Let $\boldsymbol{\beta}^*$ be an optimal solution to Problem (P-Primal), and $(\boldsymbol{\eta}^*, \zeta^*)$ be an optimal solution to Problem (P-Dual). Then we have:*

$$a_i\left(\boldsymbol{\eta}^{*\top} \boldsymbol{v}_i - \zeta^*\right) < \lambda s \quad \Rightarrow \quad \beta_i^* = 0,$$
$$a_i\left(\boldsymbol{\eta}^{*\top} \boldsymbol{v}_i - \zeta^*\right) > \lambda s \quad \Rightarrow \quad \beta_i^* = \frac{1}{s}. \tag{11}$$

The following corollary concludes this section.

**Corollary 1.** *There exist $\boldsymbol{\theta} \in \mathbb{R}^n$ and $b \in \mathbb{R}$ such that:*

$$a_i\left(\boldsymbol{\theta}^\top \boldsymbol{v}_i + b\right) < 1 \quad \Rightarrow \quad \beta_i^* = 0,$$
$$a_i\left(\boldsymbol{\theta}^\top \boldsymbol{v}_i + b\right) > 1 \quad \Rightarrow \quad \beta_i^* = \frac{1}{s}. \tag{12}$$

Corollary 1 reveals the existence of a scoring function of the form:

$$s_i \triangleq a_i\left(\boldsymbol{\theta}^\top \boldsymbol{v}_i + b\right). \tag{13}$$

As illustrated in Figure 2(a), this scoring function can precisely identify the active and inactivate variables in (P1). Tokens with low scores will be inactivate in (P1). Therefore, these tokens can be safely evicted from the cache.

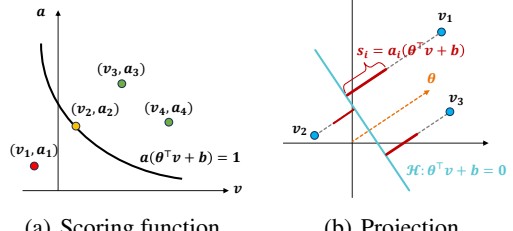

(a) Scoring function      (b) Projection

Figure 2: **Illustration of the scoring function.** (a) The scoring function $s_i$, as a function of both $\boldsymbol{v}_i$ and $a_i$, can distinguish between activate (green plots) and inactivate (red plots) samples. (b) The computation of $s_i = a_i(\boldsymbol{\theta}^\top \boldsymbol{v}_i + b)$, which represents the signed distance of $\boldsymbol{v}_i$ to the hyperplane $\mathcal{H}$, scaled by $a_i$.

**A2:** *We can define a scoring function in the form of Equation (13).*

**Remark 2.** *In Theorem 2, we have shown that $\boldsymbol{\eta}^*$ and $\zeta^*$ correspond to the parameters $\boldsymbol{\theta}$ and $b$ that we are interested in. However, it is infeasible to directly derive $\boldsymbol{\eta}^*$ and $\zeta^*$ without explicitly solving Problem (P-Dual). Therefore, we turn to an intuitive analysis on $\boldsymbol{\theta}$ and $b$ to derive a more practical form in the subsequent section.*

### 3.3 Anchor Direction Projection: Practical Form

The theoretical analysis in Section 3.2 motivates us to define a scoring function in the form of Equation 13. Despite the theoretical soundness, the determination of the parameters $\boldsymbol{\theta}$ and $b$ remains challenging, because the dual variables $\boldsymbol{\eta}^*$ and $\zeta^*$ are inaccessible before solving Problem (P-Dual). Therefore, this section aims to answer:

**Q3:** *How can we determine generally good parameters $\boldsymbol{\theta}$ and $b$ in practice?*

We begin by analyzing a special case when $\boldsymbol{\theta} = \mathbf{0}$. In this case, the scoring function is simplified to $s_i = a_i b$, where $b$ is a constant. Setting $b = 1$ gives us the most commonly used score, i.e., the attention weights. From this perspective, existing methods that relied solely on attention weights can be viewed as a special instance of our framework.

For the more general case when $\boldsymbol{\theta} \neq \mathbf{0}$, without loss of generality, we can assume that $\|\boldsymbol{\theta}\|_2 = 1$. With this assumption, the equation $\mathcal{H} : \boldsymbol{\theta}^\top \boldsymbol{x} + b = 0$ represents a hyperplane in $\mathbb{R}^n$, and $\boldsymbol{\theta}^\top \boldsymbol{v}_i + b$ represents the signed distance (a distance metric that can assume negative values) of the point $\boldsymbol{v}_i$ from $\mathcal{H}$. The factor $a_i$ scales these signed distances. This computation is illustrated in Figure 2(b).

When applying this scoring function for top-$k$ token selection, the tokens with relatively larger signed distances from $\mathcal{H}$, scaled by $a_i$, are retained, while the other tokens are discarded. This procedure essentially preserves the projection of the output $\hat{\boldsymbol{y}}$ onto the direction of $\boldsymbol{\theta}$. We refer to the direction of $\boldsymbol{\theta}$ as an *"anchor direction"*, as it acts as a reference to guide the selection of retained tokens. A natural and intuitive way is to set the direction of $\boldsymbol{y}$ as the anchor direction. This choice helps strengthen the semantic information along the direction of the pre-eviction output, ensuring that the most significant semantic features are preserved.

In this case, the proposed scoring function is defined as $s_i = a_i(\boldsymbol{y}^\top \boldsymbol{v}_i + b)$, where $b$ acts as a bias term to balance the relative importance of projections and attention weights. As $b \to \infty$, the scoring function gradually converges to the traditional attention-based score. The results suggest that a carefully chosen $b$ leads to a slight performance gain, while setting $b = 0$ also results in stable performance. Therefore, we set $b = 0$ and thus $s_i = a_i \boldsymbol{y}^\top \boldsymbol{v}_i$ in the main experiments. We conduct experiments on the choices of $\boldsymbol{\theta}$ and $b$ in Section 4.5.

**A3:** *We can simply set $\boldsymbol{\theta} = \boldsymbol{y}$ and $b = 0$.*

### 3.4 Implementation Details

In this section, we supplement some implementation details of AnDPro. Following SnapKV [16], we retain an observation window $\mathcal{W}$ to accumulate attention weights from the last several prompt tokens, which generates queries $\{\boldsymbol{q}^t\}_{t \in \mathcal{W}}$. Specifically, We retain the tokens within this window as well as the first token. At each time step in the observation window, the attention for the $i^{\text{th}}$ token and the corresponding pre-eviction output are computed as:

$$a_i^t = \underset{1 \leq i \leq t}{\text{Softmax}}(\boldsymbol{q}^{t\top} \boldsymbol{k}_i) \quad \text{and} \quad \boldsymbol{y}^t = \sum_{i=1}^n a_i^t \boldsymbol{v}_i. \tag{14}$$

The scoring function for each token is then computed as:

$$s_i \triangleq \sum_{t \in \mathcal{W}} s_i^t = \sum_{t \in \mathcal{W}} a_i^t \boldsymbol{y}^{t\top} \boldsymbol{v}_i. \tag{15}$$

We perform top-$k$ selection across all attention heads within each layer, based on their scores. This allows for flexible budget allocation among heads, significantly enhancing generation quality. Additionally, we introduce a useful technique of merging tokens into chunks. Notice that both SnapKV and Ada-KV employ max/mean pooling operations, ensuring that adjacent tokens have similar scores. This leads to a successive selection of tokens, significantly improving the performance.

Table 1: **Comparison based on Mistral-7B-Instruct-v0.2 among 16 datasets from LongBench [53].** The results for H2O, Streaming, SnapKV, and Pyramid are from Feng et al. [17], and the results for Ada-KV and CriticalKV are reproduced based on their papers [17, 45]. We mark the best results in **bold red**, and the second best results in underlined blue.

| | Single-Doc. QA | | | Multi-Doc. QA | | | Summarization | | | Few-shotLearning | | | Synthetic | | Code | | Ave. Score |
|---|---|---|---|---|---|---|---|---|---|---|---|---|---|---|---|---|---|
| | NrtvQA | Qasper | MF-en | HotpotQA | 2WikiMQA | Musique | GovReport | QMSum | MultiNews | TREC | TriviaQA | SAMSum | PCount | PRe | Lcc | RB-P | |
| Full Cache | 26.63 | 32.99 | 49.34 | 42.77 | 27.35 | 18.77 | 32.87 | 24.24 | 27.10 | 71.00 | 86.23 | 42.96 | 2.75 | 86.98 | 55.33 | 52.87 | 42.51 |
| Budget=128 | | | | | | | | | | | | | | | | | |
| H2O | 21.19 | 21.66 | 38.60 | 30.63 | 20.65 | 12.19 | 20.65 | 22.42 | 21.81 | 39.00 | 82.52 | 40.68 | 2.98 | 79.56 | 49.13 | 46.76 | 34.40 |
| Streaming | 16.61 | 14.74 | 31.40 | 28.05 | 21.36 | 12.08 | 18.44 | 18.91 | 19.26 | 43.50 | 74.22 | 29.00 | 2.75 | 31.65 | 41.27 | 38.84 | 27.63 |
| Snapkv | 19.17 | 21.40 | 42.93 | 36.76 | 22.44 | 15.86 | 19.16 | 21.84 | 21.55 | 47.50 | 84.15 | 40.24 | 2.30 | 68.26 | 50.69 | 47.13 | 35.09 |
| Pyramid | 20.16 | 21.77 | 43.55 | 36.78 | 23.12 | 14.39 | 19.53 | 22.03 | 21.47 | 51.00 | 84.62 | 40.24 | 2.79 | 70.77 | 50.57 | 46.53 | 35.58 |
| Ada-KV | 21.79 | 23.03 | 47.07 | 38.70 | 22.85 | 15.92 | 19.94 | 23.05 | 21.84 | 63.00 | 85.36 | 40.01 | 3.20 | 72.13 | 51.36 | 49.29 | 37.41 |
| CriticalKV | 21.55 | 23.27 | 46.67 | 38.95 | 24.95 | 16.85 | 20.83 | 23.14 | 22.16 | 65.00 | 85.53 | 40.02 | 3.48 | 77.81 | 53.58 | 50.98 | 38.42 |
| AnDPro | 24.84 | 25.14 | 47.33 | 39.79 | 23.99 | 17.34 | 20.40 | 23.45 | 21.94 | 67.50 | 85.59 | 40.62 | 3.19 | 76.94 | 53.77 | 51.55 | 38.96 |
| Budget=256 | | | | | | | | | | | | | | | | | |
| H2O | 21.54 | 22.92 | 42.56 | 31.07 | 22.53 | 13.76 | 22.52 | 22.40 | 23.09 | 40.50 | 84.20 | 40.77 | 3.41 | 86.10 | 50.98 | 48.17 | 36.03 |
| Streaming | 17.93 | 16.01 | 33.36 | 30.71 | 21.30 | 10.08 | 20.66 | 19.47 | 22.89 | 53.50 | 73.59 | 29.22 | 3.00 | 27.77 | 42.30 | 39.87 | 28.85 |
| SnapKV | 22.37 | 23.74 | 48.13 | 38.56 | 22.43 | 15.66 | 21.91 | 23.13 | 23.15 | 61.50 | 85.45 | 41.42 | 3.09 | 84.54 | 53.22 | 50.24 | 38.66 |
| Pyramid | 20.09 | 24.00 | 47.33 | 38.24 | 22.48 | 16.02 | 21.40 | 22.45 | 22.63 | 63.00 | 84.93 | 40.98 | 3.40 | 82.48 | 52.78 | 49.36 | 38.22 |
| Ada-KV | 23.02 | 25.70 | 49.23 | 40.18 | 24.76 | 17.43 | 22.17 | 23.21 | 23.48 | 67.50 | 85.58 | 41.67 | 2.99 | 84.65 | 55.07 | 52.17 | 40.03 |
| CriticalKV | 23.70 | 26.66 | 49.00 | 40.05 | 25.12 | 17.86 | 21.99 | 23.35 | 23.45 | 68.00 | 86.19 | 42.07 | 2.92 | 85.81 | 53.81 | 50.42 | 40.03 |
| AnDPro | 25.71 | 28.17 | 49.31 | 40.04 | 25.81 | 18.68 | 21.66 | 23.64 | 23.63 | 70.50 | 86.59 | 42.60 | 2.86 | 85.62 | 55.74 | 52.87 | 40.84 |
| Budget=512 | | | | | | | | | | | | | | | | | |
| H2O | 21.72 | 26.03 | 44.81 | 32.33 | 23.16 | 14.86 | 23.65 | 22.84 | 24.70 | 42.00 | 85.22 | 41.57 | 3.40 | 86.45 | 53.04 | 49.68 | 37.22 |
| Streaming | 18.76 | 17.17 | 37.09 | 30.21 | 21.64 | 9.93 | 24.44 | 20.00 | 25.57 | 62.00 | 72.36 | 29.95 | 2.48 | 18.17 | 43.70 | 40.13 | 29.60 |
| SnapKV | 24.60 | 27.81 | 48.98 | 39.46 | 25.25 | 16.98 | 23.70 | 22.96 | 24.37 | 67.00 | 85.88 | 41.26 | 3.16 | 85.67 | 54.16 | 51.71 | 40.26 |
| Pyramid | 23.23 | 27.94 | 48.87 | 40.50 | 24.36 | 16.74 | 23.22 | 23.16 | 24.37 | 67.00 | 85.73 | 41.74 | 3.16 | 85.67 | 54.16 | 50.34 | 40.01 |
| Ada-KV | 23.88 | 29.05 | 48.79 | 40.44 | 25.30 | 17.66 | 23.47 | 23.47 | 24.48 | 69.50 | 86.39 | 41.38 | 2.94 | 87.56 | 56.58 | 52.88 | 40.86 |
| CriticalKV | 23.83 | 28.67 | 49.13 | 40.18 | 26.18 | 18.19 | 23.52 | 23.57 | 24.28 | 67.50 | 86.39 | 42.16 | 2.82 | 87.55 | 56.51 | 52.64 | 40.98 |
| AnDPro | 24.87 | 30.99 | 49.58 | 41.34 | 27.01 | 18.44 | 23.40 | 23.76 | 25.01 | 71.00 | 86.74 | 43.65 | 2.83 | 86.23 | 57.46 | 53.61 | 41.62 |
| Budget=1024 | | | | | | | | | | | | | | | | | |
| H2O | 23.90 | 28.62 | 46.46 | 37.03 | 24.74 | 15.04 | 25.30 | 23.11 | 25.92 | 46.00 | 85.93 | 41.80 | 3.24 | 86.57 | 54.46 | 51.01 | 38.70 |
| Streaming | 19.42 | 21.69 | 41.75 | 32.40 | 22.18 | 11.18 | 27.13 | 21.09 | 26.59 | 67.00 | 71.79 | 30.11 | 2.88 | 16.57 | 44.82 | 39.76 | 31.02 |
| SnapKV | 25.47 | 29.57 | 49.33 | 40.90 | 25.53 | 19.01 | 25.94 | 23.89 | 26.21 | 69.50 | 86.48 | 42.10 | 2.98 | 88.56 | 55.57 | 51.92 | 41.44 |
| Pyramid | 24.21 | 29.86 | 48.93 | 40.75 | 25.05 | 18.77 | 25.73 | 24.06 | 25.65 | 68.50 | 86.31 | 42.25 | 2.97 | 87.17 | 54.75 | 52.10 | 41.07 |
| Ada-KV | 25.60 | 31.15 | 49.02 | 41.47 | 27.07 | 18.77 | 25.40 | 23.84 | 25.92 | 70.50 | 86.55 | 43.07 | 2.54 | 87.14 | 57.27 | 53.85 | 41.82 |
| CriticalKV | 26.03 | 31.50 | 48.39 | 41.73 | 27.10 | 18.84 | 25.30 | 24.10 | 25.99 | 71.00 | 86.30 | 43.16 | 2.66 | 87.23 | 56.69 | 53.20 | 41.83 |
| AnDPro | 25.78 | 32.54 | 49.35 | 42.20 | 26.94 | 18.63 | 25.33 | 24.01 | 26.29 | 71.00 | 86.30 | 43.46 | 2.68 | 86.98 | 57.18 | 53.77 | 42.03 |

Inspired by this, we merge tokens into chunks, each containing a predefined number of tokens, to maintain semantic coherence. The chunk size is treated as a hyperparameter. The value state of a merged chunk is the weighted sum of the original tokens' values, and its attention score is the sum of the individual attention weights. The pseudocode of AnDPro is provided in Appendix B.1.

# 4 Experiments

## 4.1 Experimental Setup

**Datasets** We conduct comprehensive experiments to demonstrate the effectiveness of our approach on two commonly-used benchmarks, Longbench [53] and Needle-in-a-Haystack [54]. Specifically, Long-Bench is a long-sequence benchmark containing 16 datasets covering multi-task domains. The Needle-in-a-Haystack dataset was specifically

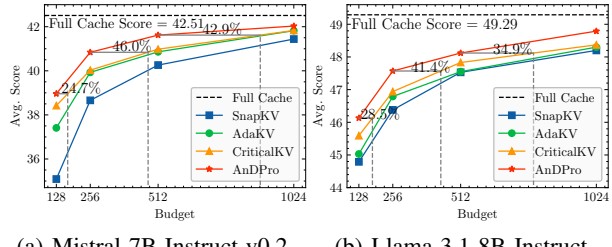

(a) Mistral-7B-Instruct-v0.2    (b) Llama-3.1-8B-Instruct

Figure 3: **Average scores on the LongBench dataset.** The numbers annotated on the gray lines represent the proportion of budget saved by our method compared to the current SOTA to achieve the same accuracy.

used to assess the context retrieval capabilities of different budget strategies, particularly in tasks involving long-context retrieval. More detailed information about these datasets are in Appendix B.2. Code is available at https://github.com/MIRALab-USTC/LLM-AnDPro.

**Baselines** We evaluate our approach using two open-source large language models: Mistral-7B-Instruct-v0.2 [55] and Llama-3.1-8B-Instruct [56]. We compare AnDPro against several strong baselines. These include representative prior works such as H2O [9], StreamingLLM [14], and SnapKV [16], as well as latest SOTA methods, including Ada-KV [17] and CriticalKV [45].

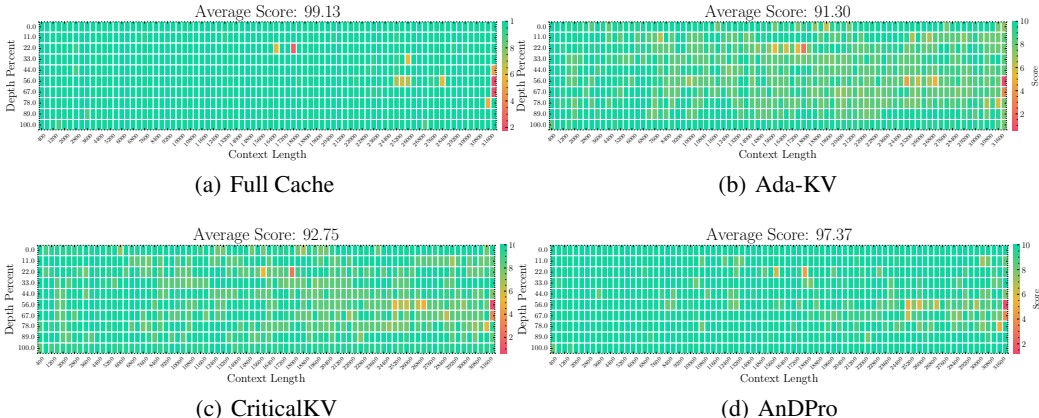

(a) Full Cache            (b) Ada-KV

(c) CriticalKV            (d) AnDPro

Figure 4: **Results of Mistral-7b-Instruct-v0.2 on the Needle-in-a-Haystack test.** The x-axis represents the document's context length, and the y-axis indicates the depth of the needle insertion. Each square in the heatmap demonstrates the correctness of retrieving a proper sentence inserted in the context for a given context length and insertion depth. The Average Score is calculated by averaging the sum of score with different insert depth across various context lengths.

**HyperParameters and Experimental Environment** In our main experiments, we evaluate the performance of different methods across a range of budget sizes, specifically setting budgets to $B \in \{128, 256, 512, 1024\}$ to test their effectiveness under different memory constraints. All experiments are run on a single $A800 - 80G$ GPU. In the main experiments, we set the observation window size to 32 and the chunk size to 4. Our implementation is built upon the Ada-KV framework, and the hyperparameters are aligned with those used in the baseline. More details can be found in Appendix B.3. We will release our code once the paper is accepted for publication.

## 4.2 Evaluations on LongBench Test

Table 1 presents the performance scores of various methods under different budget allocations for the Mistral model. Due to space limitations, results for the Llama model are provided in Appendix C.1. Results on the Chinese datasets in LongBench are in Appendix C.2. Overall, AnDPro consistently delivers state-of-the-art (SOTA) performance across both models and a range of budget configurations.

We further visualize the performance scores in Figure 3. Notably, for the Mistral model with a budget of 256 ($k = 256$), AnDPro achieves an average score of $40.84$, which is $96.07\%$ of the full cache accuracy, while utilizing only $3.44\%$ ($256/7425$) of the total KV cache budget. In comparison, the previous SOTA method, CriticalKV, requires a budget of $474$ to achieve similar performance. AnDPro achieves a $46.0\%$ reduction in budget size compared with CriticalKV. Moreover, on multiple datasets, such as MF-en, SAMSum, and Lcc, AnDPro achieves comparable or even better results compared with full cache.

## 4.3 Evaluations on Needle-in-a-Haystack Test

We evaluate the long-context retrieval capability of different KV methods using the Needle-in-a-Haystack test. This test involves retrieving a specific statement (the "needle") placed within a long context window (the "haystack"). In our experiment, we set the budget size to $k = 128$ and compare AnDPro against Ada-KV and CriticalKV. As shown in Figure 4, AnDPro significantly outperforms the baselines, achieving a score of $97.37$, which is very close to the performance of the full cache. Additional results of other baselines are in Appendix C.3.

## 4.4 Memory & Latency

We evaluate the computational efficiency of various KV eviction strategies by measuring the peak memory usage and decoding latency across different context lengths, ranging from 8K to 256K.

We conduct the experiments on the Mistral-7b-Instruct-v0.2 model with a fixed budget size $k = 128$. To measure decoding latency, we use different methods to generate 1K tokens and compute the per-token decoding latency. The results are in Figure 5. The peak memory usage and decoding latency are roughly the same for these eviction methods, significantly optimized compared with the full cache model. This demonstrates that AnDPro achieves a similar memory and

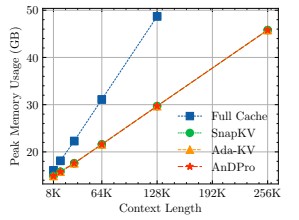 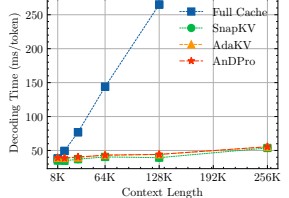

(a) Peak memory usage      (b) Decoding latency

Figure 5: **Memory and latency of different methods.**

time efficiency with much better accuracy compared with previous SOTA eviction methods. Further analysis of the inference time composition of different methods is in Appendix C.4, demonstrating that the additional computation introduced by AnDPro—specifically the Update KV Phase—incurs negligible time overhead.

## 4.5 Analysis

**Ablation Study**    We conduct ablation studies in Appendix C.5 to isolate the impact of our core contribution, i.e., the projection-based scoring function compared to attention-based scores. We then analyze the effects of three design choices in our algorithm: (1) merging tokens into chunks, (2) preserving the first token, and (3) cross-head budget allocation. The results confirm that the projection-based scoring function consistently outperforms its attention-based counterparts, and that each component contributes positively to performance in our framework.

**Different Choices of $\theta$ and $b$**    We explore different choices of anchor direction $\theta$ and the bias term $b$. The results are in Appendix C.6, demonstrating that our setting $\theta = y$ and $b = 0$ achieves a superior and robust performance.

**Eviction Loss and Score Visualization**    We analyze the eviction loss between pre- and post-eviction outputs. Results in Appendix C.7 demonstrate that our method can effectively reduce this loss. Furthermore, we visualize the distribution of token scores in Appendix C.8.

**Value Vector Analysis**    We investigate the role of value vectors in the eviction process. Using PCA, we project value vectors and model outputs into two-dimensional space. The results are in Appendix C.9. We also provide statistical histograms of value vector magnitudes in Figure 15 in Appendix C.9. These results help understand the reason why projection-based score works intuitively.

**Case Study**    We provide some case studies in Appendix C.10, showing that AnDPro can more accurately preserve important tokens compared to attention-based scoring functions.

**Longer context**    we evaluate performance on extremely long contexts ranging from 64k to 384k tokens. The results are in Appendix C.11, demonstrating that AnDPro achieves the best performance even on much longer contexts.

## 5 Conclusion

In this work, we model KV cache eviction as a combinatorial optimization problem, and relaxes it to a sparse optimization problem. Motivated by the theoretical and intuitive analysis, we proposes **AnDPro** (**An**chor **D**irection **Pro**jection), which leverages projection-based scoring functions for more accurate KV cache eviction. Extensive experiments on LongBench and Needle-in-a-Haystack benchmarks demonstrate the effectiveness of our approach.

## Acknowledgements

The authors would like to thank all the anonymous reviewers for their insightful comments. This work was supported in part by National Key R&D Program of China under contract 2022ZD0119801, National Nature Science Foundations of China grants U23A20388 and 62021001.

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

# A  Proofs

**Proposition 1.** *Let $S \subset [n]$ be the set of retained tokens, and define the indicator function*

$$z_i \triangleq \begin{cases} 1, & i \in S, \\ 0, & i \notin S \end{cases} \tag{5}$$

*to represent whether the $i^{th}$ token is retained. Then, the post-eviction output can be expressed as:*

$$\hat{\boldsymbol{y}} = \frac{\sum_{i=1}^{n} z_i a_i \boldsymbol{v}_i}{\sum_{i=1}^{n} z_i a_i}. \tag{6}$$

*Proof.* We have

$$
\begin{aligned}
\hat{\boldsymbol{y}} &= \sum_{i \in S} \operatorname*{Softmax}_{i \in S}(\boldsymbol{q}_n^\top \boldsymbol{k}_i)\boldsymbol{v}_i = \sum_{i \in S} \frac{\exp(\boldsymbol{q}_n^\top \boldsymbol{k}_i)}{\sum_{j \in S} \exp(\boldsymbol{q}_n^\top \boldsymbol{k}_j)} \boldsymbol{v}_i \\
&= \frac{\sum_{i=1}^{n} z_i \frac{\exp(\boldsymbol{q}_n^\top \boldsymbol{k}_i)}{\sum_{k=1}^{n} \exp(\boldsymbol{q}_n^\top \boldsymbol{k}_k)} \boldsymbol{v}_i}{\sum_{j=1}^{n} z_i \frac{\exp(\boldsymbol{q}_n^\top \boldsymbol{k}_j)}{\sum_{k=1}^{n} \exp(\boldsymbol{q}_n^\top \boldsymbol{k}_k)}} = \frac{\sum_{i=1}^{n} z_i a_i \boldsymbol{v_i}}{\sum_{i=1}^{n} z_i a_i}.
\end{aligned} \tag{16}
$$

$\square$

**Theorem 1.** *The dual problem to Problem (P-Primal) is given by:*

$$\sup_{\boldsymbol{\eta},\zeta} -\frac{1}{2}\|\boldsymbol{\eta}\|^2 + \boldsymbol{\eta}^\top \boldsymbol{y} - \zeta + \frac{1}{s}\sum_{i=1}^{n} \rho_i(\boldsymbol{\eta}, \zeta), \tag{P-Dual}$$

*where $\boldsymbol{\eta}^* \in \mathbb{R}^n$ and $\zeta \in \mathbb{R}$ are dual variables, and*

$$\rho_i(\boldsymbol{\eta}, \zeta) \triangleq \min\left(0, \lambda s - a_i\left(\boldsymbol{\eta}^\top \boldsymbol{v}_i - \zeta\right)\right). \tag{10}$$

*Proof.* Recall that Problem (P-Primal) takes the form of:

$$\min_{\boldsymbol{\beta} \in [0,\frac{1}{s}]^n} \quad \frac{1}{2}\|\boldsymbol{u}\|_2^2 + \lambda s \sum_{i=1}^{n} \beta_i$$

$$\text{s.t.} \quad \boldsymbol{u} = \boldsymbol{y} - \sum_{i=1}^{n} \beta_i a_i \boldsymbol{v}_i$$

$$\sum_{i=1}^{n} \beta_i a_i = 1$$

By introducing the new variables $\boldsymbol{\eta} \in \mathbb{R}^n$ and $\zeta \in \mathbb{R}$, we get its Lagragian:

$$L(\boldsymbol{\beta}, \boldsymbol{u}, \boldsymbol{\eta}, \zeta) = \frac{1}{2}\|\boldsymbol{u}\|_2^2 + \lambda s \sum_{i=1}^{n} \beta_i + \boldsymbol{\eta}^\top \left(\boldsymbol{y} - \sum_{i=1}^{n} \beta_i a_i \boldsymbol{v}_i - \boldsymbol{u}\right) + \zeta\left(\sum_{i=1}^{n} \beta_i a_i - 1\right). \tag{17}$$

The primal variables are $\beta \in [0, \frac{1}{s}]^n$ and $\boldsymbol{u} \in \mathbb{R}^n$. The dual function $g(\boldsymbol{\eta}, \zeta)$ is then:

$$
\begin{aligned}
g(\boldsymbol{\eta}, \zeta) &= \inf_{\boldsymbol{\beta}, \boldsymbol{u}} L(\boldsymbol{\beta}, \boldsymbol{u}, \boldsymbol{\eta}, \zeta) \\
&= \boldsymbol{\eta}^\top \boldsymbol{y} - \zeta + \inf_{\boldsymbol{u}}\left(\frac{1}{2}\|\boldsymbol{u}\|^2 - \boldsymbol{\eta}^\top \boldsymbol{u}\right) + \inf_{\boldsymbol{\beta} \in [0,\frac{1}{s}]^n}\left(\lambda s \sum_{i=1}^{n} \beta_i - \sum_{i=1}^{n} \beta_i a_i \boldsymbol{\eta}^\top \boldsymbol{v}_i + \zeta \sum_{i=1}^{n} \beta_i a_i\right) \\
&= \boldsymbol{\eta}^\top \boldsymbol{y} - \zeta + \inf_{\boldsymbol{u}}\left(\frac{1}{2}\|\boldsymbol{u}\|^2 - \boldsymbol{\eta}^\top \boldsymbol{u}\right) + \sum_{i=1}^{n} \inf_{\beta_i \in [0,\frac{1}{s}]} \beta_i\left(\lambda s - a_i\left(\boldsymbol{\eta}^\top \boldsymbol{v}_i - \zeta\right)\right).
\end{aligned} \tag{18}
$$

It is easy to derive that

$$\inf_{\boldsymbol{u}}\left(\frac{1}{2}\|\boldsymbol{u}\|^2 - \boldsymbol{\eta}^\top \boldsymbol{u}\right) = -\frac{1}{2}\|\boldsymbol{\eta}\|^2, \tag{19}$$

with the optimal solution $\boldsymbol{u}^* = \boldsymbol{\eta}$, and

$$\inf_{\beta_i \in [0, \frac{1}{s}]} \beta_i \left( \lambda s - a_i \left( \boldsymbol{\eta}^\top \boldsymbol{v}_i - \zeta \right) \right) = \frac{1}{s} \min \left( 0, \lambda s - a_i \left( \boldsymbol{\eta}^\top \boldsymbol{v}_i - \zeta \right) \right). \tag{20}$$

with the optimal solutions

$$\beta_i^* = \begin{cases} 0, & \lambda s - a_i \left( \boldsymbol{\eta}^\top \boldsymbol{v}_i - \zeta \right) > 0, \\ \in \left[ 0, \frac{1}{s} \right], & \lambda s - a_i \left( \boldsymbol{\eta}^\top \boldsymbol{v}_i - \zeta \right) = 0, \\ \frac{1}{s}, & \lambda s - a_i \left( \boldsymbol{\eta}^\top \boldsymbol{v}_i - \zeta \right) < 0. \end{cases} \tag{21}$$

Therefore, we have:

$$g(\boldsymbol{\eta}, \zeta) = -\frac{1}{2} \|\boldsymbol{\eta}\|^2 + \boldsymbol{\eta}^\top \boldsymbol{y} - \zeta + \frac{1}{s} \sum_{i=1}^{n} \min \left( 0, \lambda - a_i \left( \boldsymbol{\eta}^\top \boldsymbol{v}_i - \zeta \right) \right). \tag{22}$$

Combining the above, we get the dual problem:

$$\sup_{\boldsymbol{\eta}, \zeta} \quad g(\boldsymbol{\eta}, \zeta) = -\frac{1}{2} \|\boldsymbol{\eta}\|^2 + \boldsymbol{\eta}^\top \boldsymbol{y} - \zeta + \frac{1}{s} \sum_{i=1}^{n} \min \left( 0, \lambda s - a_i \left( \boldsymbol{\eta}^\top \boldsymbol{v}_i - \zeta \right) \right). \tag{23}$$

$\square$

**Theorem 2.** *Let $\boldsymbol{\beta}^*$ be an optimal solution to Problem (P-Primal), and $(\boldsymbol{\eta}^*, \zeta^*)$ be an optimal solution to Problem (P-Dual). Then we have:*

$$\begin{aligned} a_i \left( \boldsymbol{\eta}^{*\top} \boldsymbol{v}_i - \zeta^* \right) < \lambda s \quad &\Rightarrow \quad \beta_i^* = 0, \\ a_i \left( \boldsymbol{\eta}^{*\top} \boldsymbol{v}_i - \zeta^* \right) > \lambda s \quad &\Rightarrow \quad \beta_i^* = \frac{1}{s}. \end{aligned} \tag{11}$$

*Proof.* By the strong duality, it is easy to show that $(\boldsymbol{\eta}^*, \zeta^*)$ is a geometric multiplier. Therefore, from the optimal condition, we have

$$(\boldsymbol{\beta}^*, \boldsymbol{u}^*) \in \inf_{\boldsymbol{\beta}, \boldsymbol{u}} L(\boldsymbol{\beta}, \boldsymbol{u}, \boldsymbol{\eta}^*, \zeta^*). \tag{24}$$

We then complete the proof according to Equation 21. $\square$

**Corollary 1.** *There exist $\boldsymbol{\theta} \in \mathbb{R}^n$ and $b \in \mathbb{R}$ such that:*

$$\begin{aligned} a_i \left( \boldsymbol{\theta}^\top \boldsymbol{v}_i + b \right) < 1 \quad &\Rightarrow \quad \beta_i^* = 0, \\ a_i \left( \boldsymbol{\theta}^\top \boldsymbol{v}_i + b \right) > 1 \quad &\Rightarrow \quad \beta_i^* = \frac{1}{s}. \end{aligned} \tag{12}$$

*Proof.* We can take

$$\boldsymbol{\theta} \triangleq \frac{\boldsymbol{\eta}^*}{\lambda s}, \quad b \triangleq -\frac{\zeta^*}{\lambda s}, \tag{25}$$

where $\boldsymbol{\eta}^*$ and $\zeta^*$ are the optimal solution to Problem (P-Dual). This completes the proof. $\square$

# B Implementation Details

## B.1 Algorithm

A simplified pseudocode of AnDPro is provided in Algorithm 1. We omit some unimportant details (such as preserving the first token) for simplicity.

---

**Algorithm 1** AnDPro Algorithm in One Layer

---

**Require:** A set $H$ of all heads, A set $\mathcal{W}$ of the observation window, Number of tokens $L$ outside the observation window, Queries $\{\boldsymbol{q}^{h,t}\}_{h\in H, t\in\mathcal{W}}$, Keys $\{\boldsymbol{k}_i^h\}_{h\in H, i\in[L]}$, Values $\{\boldsymbol{v}_i^h\}_{h\in H, i\in[L]}$, Chunk size $C$, Budget size $B$ for each head
**Ensure:** Retained KV cache $\{S^h\}_{h\in H}$
1: $\boldsymbol{a}_i^{h,t}\,|_{h\in H, t\in\mathcal{W}, i\in[L]}\leftarrow \mathrm{Softmax}_{i\in[L]}(\boldsymbol{q}^{h,t}\boldsymbol{v}_i^h)$
2: $\hat{\boldsymbol{v}}_i^{h,t}\,|_{h\in H, t\in\mathcal{W}, i\in[L/C]}\leftarrow \sum_{j\in\mathrm{Chunk}_i} a_j^{h,t}\boldsymbol{v}_j^h$
3: $\hat{a}_i^{h,t}\,|_{h\in H, t\in\mathcal{W}, i\in[L/C]}\leftarrow \sum_{j\in\mathrm{Chunk}_i} a_j^{h,t}$
4: $\hat{\mathcal{W}}\leftarrow$ observation window of chunks
5: $\boldsymbol{y}^{h,t}\,|_{h\in H, t\in\hat{\mathcal{W}}}\leftarrow \sum_{i\in[L/C]}\hat{\boldsymbol{v}}_i^{h,t}$
6: $\boldsymbol{s}_i^{h,t}\,|_{h\in H, t\in\hat{\mathcal{W}}, i\in[L/C]}\leftarrow \langle\boldsymbol{y}^{h,t}, \hat{\boldsymbol{v}}_i^{h,t}\rangle$
7: $\boldsymbol{s}_i^h\,|_{h\in H, i\in[L/C]}\leftarrow \sum_{t\in\hat{\mathcal{W}}}\boldsymbol{s}_i^{h,t}$
8: $\boldsymbol{s}\leftarrow \mathrm{Concat}_{h\in H, i\in[L/C]}(\boldsymbol{s}_i^h)$
9: $\{S^h\}_{h\in H}\leftarrow \mathrm{TopK}(\boldsymbol{s}, k=|H|\times B)$
10: $S^h\,|_{h\in H}\leftarrow S^h\cup\mathcal{W}$
11: **return** $\{S^h\}_{h\in H}$

---

## B.2 Datasets

Longbench is a comprehensive benchmark that consists of 16 datasets and serves as a robust evaluation tool due to its wide variety of prompts with different domains, types, and lengths. It is designed for long-sequence tasks spanning multiple domains, including single-document QA, multi-document QA, summarization, few-shot learning, synthetic tasks, and code generation. Table 2 provides detailed information of the 16 datasets in LongBench. These datasets have average input lengths ranging from $1,235$ to $18,409$ tokens, with an average token length of $7,425$. For our evaluation, we followed the recommended evaluation procedures for each dataset, with scores ranging from $0$ to $100$.

Table 2: Details of 16 datasets from the LongBench benchmark.

| Task | Task Type | Eval Metric | Avg Len | Language | Sample Num |
|------|-----------|-------------|---------|----------|------------|
| NarrativeQA | Single-Doc. QA | F1 | 18,409 | EN | 200 |
| Qasper | Single-Doc. QA | F1 | 3,619 | EN | 200 |
| MultiFieldQA-en | Single-Doc. QA | F1 | 4,559 | EN | 150 |
| HotpotQA | Multi-Doc. QA | F1 | 9,151 | EN | 200 |
| 2WikiMultihopQA | Multi-Doc. QA | F1 | 4,887 | EN | 200 |
| MuSiQue | Multi-Doc. QA | F1 | 11,214 | EN | 200 |
| GovReport | Summarization | Rouge-L | 8,734 | EN | 200 |
| QMSum | Summarization | Rouge-L | 10,614 | EN | 200 |
| MultiNews | Summarization | Rouge-L | 2,113 | EN | 200 |
| TREC | Few-shot Learning | Accuracy | 5,177 | EN | 200 |
| TriviaQA | Few-shot Learning | F1 | 8,209 | EN | 200 |
| SAMSum | Few-shot Learning | Rouge-L | 6,258 | EN | 200 |
| PassageCount | Synthetic | Accuracy | 11,141 | EN | 200 |
| PassageRetrieval-en | Synthetic | Accuracy | 9,289 | EN | 200 |
| LCC | Code Edit | Sim | 1,235 | Python/C#/Java | 500 |
| RepoBench-P | Code Edit | Sim | 4,206 | Python/Java | 500 |

### B.3   Code and Hyperparameter

Our code framework is adapted from Ada-KV [17], specifically leveraging its concise and user-friendly custom classes to facilitate more efficient and convenient budget filtering and allocation. The primary updates include a cross-head filtering strategy. We employ max pooling with a kernel size of 7 and set the size of observation window to 32. The code is released at `https://github.com/MIRALab-USTC/LLM-AnDPro`.

# C  Additional Results

## C.1  Main results of Llama on LongBench

Table 3 shows the scores of different methods based on the Llama model in 16 datasets. Overall, the results are consistent with those of Mistral, and AnDPro also leads to improved quality after cache eviction.

Table 3: **Comparison based on Llama-3.1-8B-Instruct among** 16 **datasets from LongBench.** We run all the baselines to derive the results based on their released code. We mark the best results in **bold red**, and we mark the second best results in underlined blue.

| | Single-Doc. QA | | | Multi-Doc. QA | | | Summarization | | | Few-shot Learning | | | Synthetic | | Code | | |
| --- | --- | --- | --- | --- | --- | --- | --- | --- | --- | --- | --- | --- | --- | --- | --- | --- | --- |
| | NrtvQA | Qasper | MF-en | HotpotQA | 2WikiMQA | Musique | GovReport | QMSum | MultiNews | TREC | TriviaQA | SAMSum | PCount | PRe | Lcc | RB-P | Ave. Score |
| Full Cache | 30.12 | 46.60 | 56.41 | 58.10 | 49.01 | 32.52 | 34.04 | 25.37 | 27.06 | 73.00 | 91.90 | 43.46 | 7.06 | 100.00 | 62.13 | 51.80 | 49.29 |
| **Budget=128** | | | | | | | | | | | | | | | | | |
| SnapKV | 25.54 | 31.40 | 50.99 | 55.17 | 45.50 | 27.88 | 20.77 | 23.41 | 21.05 | 47.50 | 90.46 | 40.76 | 8.00 | 99.50 | 56.88 | 46.50 | 43.21 |
| PyramidKV | 27.83 | 32.98 | 51.45 | 56.72 | 44.67 | 30.54 | 21.89 | 22.90 | 21.67 | 63.00 | 90.45 | 40.06 | 8.00 | 100.00 | 55.37 | 43.79 | 44.46 |
| Ada-KV | 25.06 | 34.50 | 52.27 | 56.29 | 47.57 | 29.06 | 21.17 | 23.36 | 21.51 | 62.00 | 92.17 | 40.84 | 8.00 | 99.50 | 58.45 | 48.72 | 45.03 |
| CriticalKV | 26.56 | 35.71 | 52.90 | 57.21 | 47.53 | 29.57 | 21.77 | 23.64 | 21.79 | 63.50 | 91.44 | 41.62 | 7.75 | 100.00 | 59.09 | 49.32 | 45.59 |
| AnDPro | 26.66 | 38.22 | 54.54 | 57.47 | 47.57 | 29.42 | 22.09 | 23.67 | 21.81 | 67.00 | 92.35 | 40.90 | 7.70 | 100.00 | 59.15 | 50.06 | 46.13 |
| **Budget=256** | | | | | | | | | | | | | | | | | |
| SnapKV | 26.92 | 38.19 | 52.62 | 57.03 | 46.75 | 29.63 | 23.07 | 23.92 | 22.82 | 58.50 | 91.83 | 40.81 | 7.75 | 99.50 | 59.95 | 48.63 | 45.50 |
| PyramidKV | 28.46 | 39.36 | 54.30 | 56.77 | 45.50 | 30.74 | 23.84 | 23.88 | 22.91 | 69.00 | 91.09 | 40.74 | 7.88 | 99.50 | 56.94 | 45.18 | 46.01 |
| Ada-KV | 27.04 | 39.24 | 54.41 | 57.36 | 47.53 | 31.35 | 23.06 | 23.81 | 23.43 | 69.00 | 92.50 | 41.23 | 7.67 | 100.00 | 61.45 | 49.52 | 46.79 |
| CriticalKV | 26.09 | 40.60 | 54.23 | 57.12 | 47.21 | 30.72 | 23.68 | 24.37 | 23.34 | 70.00 | 92.75 | 42.11 | 7.67 | 100.00 | 61.70 | 49.46 | 46.94 |
| AnDPro | 27.28 | 42.09 | 54.72 | 57.98 | 47.52 | 31.98 | 23.57 | 24.27 | 23.58 | 72.00 | 92.54 | 41.55 | 7.67 | 100.00 | 62.57 | 51.77 | 47.57 |
| **Budget=512** | | | | | | | | | | | | | | | | | |
| SnapKV | 28.40 | 41.52 | 55.40 | 57.66 | 47.48 | 30.95 | 24.92 | 24.46 | 24.54 | 68.50 | 92.33 | 41.98 | 7.33 | 99.50 | 61.25 | 50.39 | 47.29 |
| PyramidKV | 29.15 | 43.04 | 55.55 | 57.71 | 47.67 | 31.27 | 26.05 | 24.08 | 24.54 | 70.50 | 92.67 | 41.52 | 7.33 | 100.00 | 59.96 | 47.51 | 47.41 |
| Ada-KV | 26.92 | 42.68 | 55.20 | 57.47 | 48.06 | 31.27 | 24.95 | 24.17 | 24.40 | 71.00 | 92.14 | 42.35 | 7.33 | 100.00 | 62.26 | 50.62 | 47.55 |
| CriticalKV | 26.75 | 43.75 | 55.45 | 58.25 | 48.11 | 32.03 | 24.96 | 24.44 | 24.50 | 72.50 | 92.14 | 41.92 | 7.12 | 100.00 | 62.25 | 51.17 | 47.83 |
| AnDPro | 28.03 | 44.25 | 55.61 | 58.29 | 48.30 | 31.99 | 24.86 | 24.55 | 24.83 | 73.00 | 91.97 | 42.08 | 7.33 | 100.00 | 62.68 | 52.12 | 48.12 |
| **Budget=1024** | | | | | | | | | | | | | | | | | |
| SnapKV | 28.32 | 45.37 | 56.81 | 58.16 | 48.28 | 31.89 | 26.91 | 24.58 | 25.82 | 70.50 | 92.22 | 43.08 | 6.96 | 100.00 | 62.46 | 51.50 | 48.30 |
| PyramidKV | 29.32 | 44.57 | 56.14 | 58.39 | 48.36 | 31.93 | 28.20 | 23.78 | 26.00 | 70.50 | 92.41 | 42.46 | 6.88 | 100.00 | 61.33 | 49.12 | 48.09 |
| Ada-KV | 28.52 | 46.01 | 55.72 | 57.85 | 48.10 | 32.02 | 26.98 | 24.72 | 25.96 | 72.00 | 91.97 | 42.55 | 6.96 | 100.00 | 61.98 | 51.52 | 48.30 |
| CriticalKV | 28.15 | 46.19 | 56.00 | 57.99 | 48.47 | 32.68 | 27.20 | 24.69 | 25.74 | 72.50 | 91.97 | 42.33 | 6.92 | 100.00 | 61.79 | 51.36 | 48.37 |
| AnDPro | 29.40 | 46.83 | 56.72 | 58.01 | 48.68 | 32.64 | 26.69 | 24.96 | 25.92 | 73.00 | 91.97 | 43.55 | 7.38 | 100.00 | 62.43 | 52.39 | 48.79 |

## C.2  Results on the Chinese Datasets in LongBench

Table 4 shows the scores of different methods based on the Llama model on 5 Chinese datasets fron LongBench. Although Llama model don't support Chinese language, AnDPro leads to improved quality after cache eviction.

Table 4: Comparison based on Llama-3.1-8B-Instruct on 5 Chinese datasets from LongBench. The results demonstrate that AnDPro still achieves the best overall performance on Chinese datasets.

| | MF-zh | dureader | vcsum | lsht | PRe-zh | Avg. |
| --- | --- | --- | --- | --- | --- | --- |
| Full Cache | 58.69 | 33.78 | 17.32 | 46.00 | 97.67 | 50.69 |
| **Budget=128** | | | | | | |
| CriticalKV | 42.57 | 24.32 | 12.51 | 29.50 | 93.75 | 40.53 |
| Ada-KV | 52.32 | 24.33 | 12.15 | 25.75 | 97.25 | 42.36 |
| AnDPro | 55.40 | 24.77 | 13.04 | 39.50 | 97.75 | 46.09 |
| **Budget=256** | | | | | | |
| CriticalKV | 48.12 | 24.74 | 13.52 | 24.00 | 94.00 | 40.88 |
| Ada-KV | 54.86 | 24.88 | 13.40 | 23.52 | 98.00 | 42.93 |
| AnDPro | 56.84 | 25.71 | 13.92 | 42.50 | 98.50 | 47.49 |
| **Budget=512** | | | | | | |
| CriticalKV | 55.51 | 26.21 | 14.47 | 29.25 | 79.66 | 41.02 |
| Ada-KV | 57.83 | 25.75 | 14.82 | 22.50 | 97.67 | 43.71 |
| AnDPro | 57.99 | 27.63 | 14.65 | 45.00 | 98.50 | 48.75 |
| **Budget=1024** | | | | | | |
| CriticalKV | 57.07 | 27.40 | 15.47 | 23.00 | 91.50 | 42.89 |
| Ada-KV | 58.54 | 27.19 | 15.47 | 22.00 | 98.00 | 44.24 |
| AnDPro | 58.57 | 29.07 | 15.85 | 45.50 | 98.17 | 49.43 |

## C.3  Results of Needle-in-a-Haystack test

Figure 6 presents the Needle-in-a-Haystack (NIAH) scores across all methods. AnDPro significantly outperforms all baselines, achieving a high score of 97.37, which is very close to the performance of the full cache.

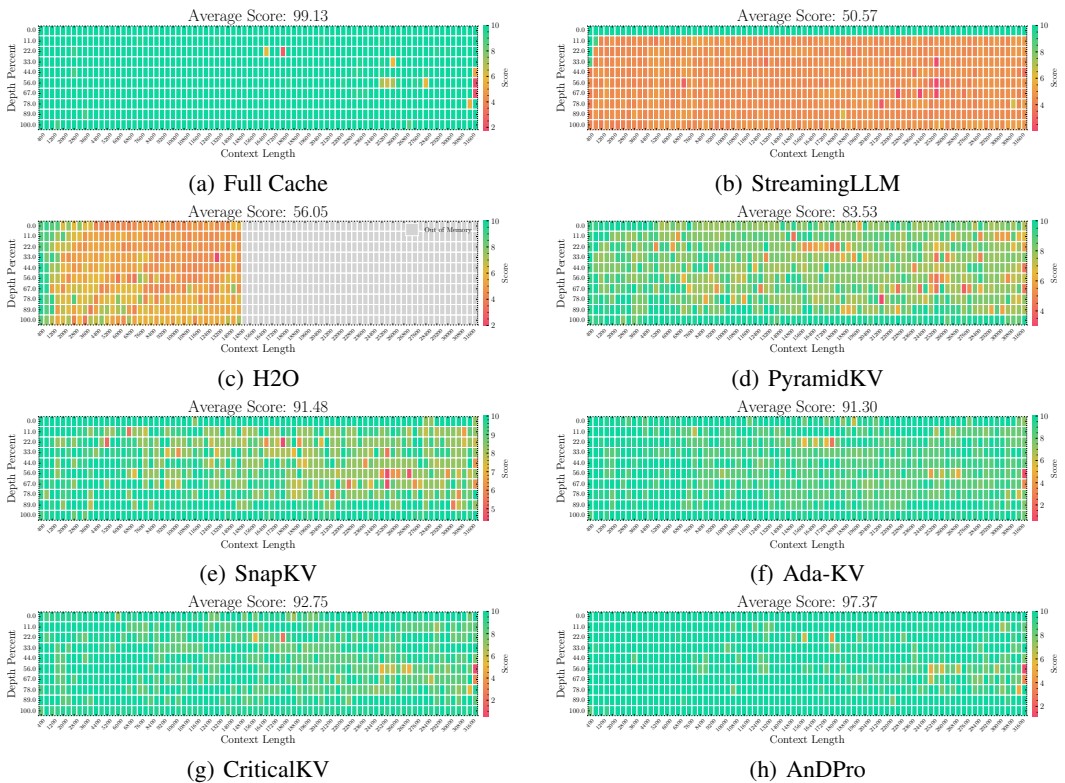

Figure 6: **Results of Mistral-7b-Instruct-v0.2 on the Needle-in-a-Haystack test.** The x-axis represents the document's context length, and the y-axis indicates the depth of the needle insertion. Each square in the heatmap demonstrates the correctness of retrieving a proper sentence inserted in the context for a given context length and insertion depth. The Average Score is calculated by averaging the sum of score with different insert depth across various context lengths.

## C.4  Runtime Analysis

We visualize The detailed comparison of time consumption across different phases in the inference process in Figure 7. The results demonstrate that the additional computation introduced by AnDPro—specifically the Update KV Phase—incurs negligible time overhead.

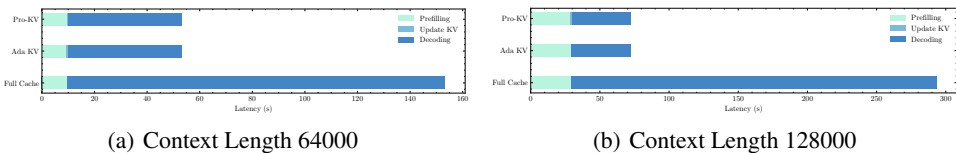

Figure 7: The detailed comparison of time consumption across different phases in the inference process under varying input lengths. The figure demonstrates the time costs of three key phases when processing input sequences of 64,000 and 128,000 tokens with an output length of 1,000 tokens.

To provide a more detailed quantitative breakdown, the specific runtime measurements are presented below (unit: seconds/milliseconds):

Table 5: Runtime analysis of different methods on sequences of length 64K and 128K.

| 64K | Prefilling (s) | KV update (s) | Layer-wise KV update (ms) |
|---|---|---|---|
| Full | 9.45 | - | - |
| SnapKV | 9.50 | 0.10 | 3.05 |
| Ada-KV | 9.88 | 0.43 | 13.28 |
| AnDPro | 10.04 | 0.58 | 18.13 |
| 128K | Prefilling (s) | KV update (s) | Layer-wise KV update (ms) |
| Full | 28.78 | - | - |
| SnapKV | 29.00 | 0.11 | 3.42 |
| Ada-KV | 29.47 | 0.59 | 18.31 |
| AnDPro | 29.78 | 0.88 | 27.41 |

Although AnDPro introduces slightly higher update latency compared to SnapKV and AdaKV, the additional overhead is negligible relative to the total prefilling time.

We further conduct experiments using the Mistral-7B-Instruct-v0.2 model on the Needle-in-a-Haystack (NIAH) benchmark. We measure the decoding latency (in seconds) for an output length of 1K under various input lengths (from 8K to 256K tokens) with a budget size of 128. The results are in Table 6.

Table 6: AnDPro maintains comparable runtime efficiency to AdaKV and remains close to SnapKV across all input lengths.

| Input Length | 8K | 16K | 32K | 64K | 128K | 256K |
|---|---|---|---|---|---|---|
| SnapKV | 35.10 | 34.72 | 37.15 | 40.49 | 39.40 | 53.69 |
| AdaKV | 39.22 | 38.99 | 39.64 | 43.39 | 44.23 | 55.30 |
| AnDPro | 39.23 | 38.67 | 40.44 | 42.98 | 44.20 | 55.51 |

## C.5 Ablation Study

### C.5.1 Ablation Study on Projection-Based Scoring Function

We conduct comprehensive ablation studies to isolate the effectiveness of the projection-based scoring function—which is the core contribution of this paper—and distinguish it from the other components in our method. Specifically, we conduct 5 sets of experiments to compare the projection-based scoring function (`ProjScore`) with the attention-based scoring function (`AttnScore`) with different combinations of other components. Specifically, we consider five experimental groups A–E:

- **A**: CrossHead + Chunk + FirstToken with `ProjScore` vs. `AttenScore`
- **B**: Chunk + FirstToken with `ProjScore` vs. Pooling + FirstToken with `AttenScore`
- **C**: CrossHead + FirstToken with `ProjScore` vs. `AttenScore`
- **D**: CrossHead + Chunk with `ProjScore` vs. CrossHead + Pooling with `AttenScore`
- **E**: Isolated `ProjScore` vs. `AttenScore` comparison

We conduct these ablation studies on the three datasets from the Single-Document QA task with the Mistral model. The results are in Table 7 and visualized in Figure 8. The results demonstrate that the projection-based scoring function consistently outperforms the attention-based scoring function, across all different configurations of other components.

We further validate this conclusion on two other based models Qwen and Llama. Table 8 and Table 9 show that AnDPro has good performance on other models, confirming both its effectiveness and robustness.

### C.5.2 Ablation Study on Other Adopted Techniques

We further conduct ablation studies to investigate the contributions of three different components in our algorithm: (1) merging tokens into chunks, (2) preserving the first token, and (3) cross-head budget allocation. The results are in Figure 9. It visually compares the performance improvements contributed by each component to the project-based method, demonstrating that all components are indispensable for achieving the final high-performance results.

Table 7: Ablation results of Mistral-7b-Instruct-v0.2 on LongBench (single QA Doc). The comparisons demonstrate that ProjScore consistently outperforms AttnScore across different settings of adopted techniques, isolating the novel contribution of AnDPro.

| | Budget | Nrtvqa | Qasper | MF-en | Avg. | Nrtvqa | Qasper | MF-en | Avg. |
|---|---|---|---|---|---|---|---|---|---|
| | | CrossHead + Chunk + FirstToken + ProjScore | | | | CrossHead + Chunk + FirstToken +AttenScore | | | |
| A | 128 | 24.84 | 25.14 | 47.33 | **32.44** | 22.19 | 23.67 | 48.33 | 31.40 |
| | 256 | 25.71 | 28.17 | 49.31 | **34.40** | 23.85 | 28.12 | 49.29 | 33.75 |
| | 512 | 24.87 | 30.99 | 49.58 | **35.15** | 25.25 | 30.21 | 48.94 | 34.80 |
| | 1024 | 25.87 | 32.54 | 49.35 | **35.92** | 25.94 | 32.45 | 49.02 | 35.80 |
| | | Chunk + FirstToken +ProjScore | | | | Pooling + FirstToken + AttenScore | | | |
| B | 128 | 20.25 | 20.92 | 46.89 | **29.35** | 19.17 | 21.40 | 42.93 | 27.83 |
| | 256 | 22.81 | 25.25 | 48.30 | **32.12** | 22.37 | 23.74 | 48.13 | 31.41 |
| | 512 | 24.72 | 27.61 | 49.38 | **33.90** | 24.60 | 27.81 | 48.98 | 33.80 |
| | 1024 | 26.09 | 30.75 | 49.20 | **35.35** | 25.47 | 29.57 | 49.33 | 34.79 |
| | | CrossHead + FirstToken +ProjScore | | | | CrossHead + FirstToken +AttenScore | | | |
| C | 128 | 22.78 | 24.98 | 47.39 | **31.72** | 21.12 | 21.16 | 38.46 | 26.91 |
| | 256 | 24.32 | 27.41 | 48.49 | **33.41** | 21.50 | 23.50 | 40.38 | 28.46 |
| | 512 | 25.64 | 29.03 | 49.09 | **34.59** | 23.93 | 24.67 | 43.08 | 30.56 |
| | 1024 | 26.30 | 30.90 | 49.08 | **35.43** | 24.94 | 27.82 | 46.60 | 33.12 |
| | | CrossHead + Chunk +ProjScore | | | | CrossHead + Pooling +AttnScore | | | |
| D | 128 | 24.90 | 24.47 | 48.57 | **32.65** | 21.64 | 23.77 | 46.48 | 30.63 |
| | 256 | 24.92 | 28.25 | 49.21 | **34.13** | 23.29 | 26.32 | 48.99 | 32.87 |
| | 512 | 26.18 | 30.20 | 49.41 | **35.26** | 24.73 | 28.59 | 48.84 | 34.05 |
| | 1024 | 26.14 | 31.57 | 49.60 | **35.77** | 25.60 | 31.27 | 48.49 | 35.12 |
| | | ProjScore | | | | AttnScore | | | |
| E | 128 | 20.27 | 22.02 | 43.67 | **28.65** | 21.50 | 20.85 | 39.11 | 27.15 |
| | 256 | 22.33 | 24.95 | 48.51 | **31.93** | 22.48 | 22.65 | 43.64 | 29.59 |
| | 512 | 24.91 | 26.80 | 49.01 | **33.57** | 23.07 | 25.91 | 44.84 | 31.27 |
| | 1024 | 25.20 | 30.02 | 49.25 | **34.82** | 25.02 | 29.48 | 47.25 | 33.92 |

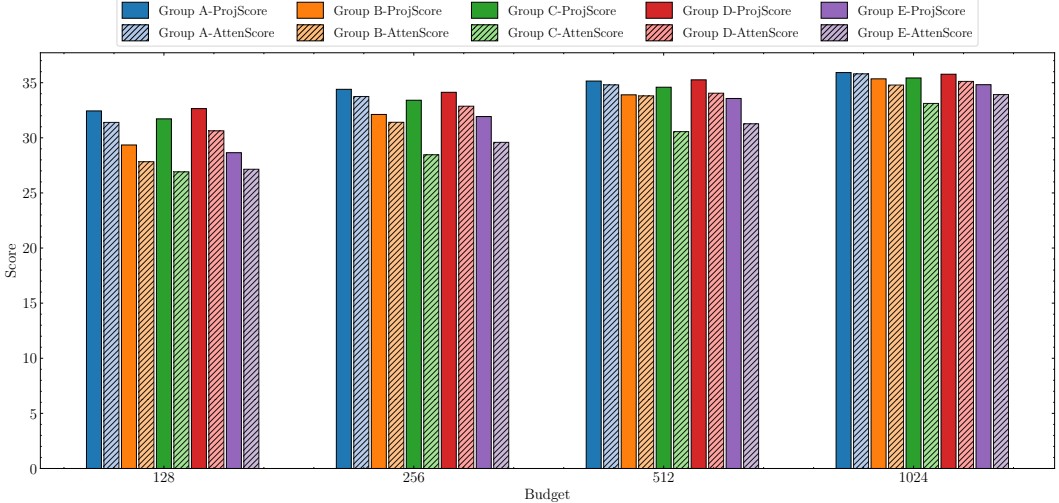

Figure 8: **Ablation results of Mistral-7b-Instruct-v0.2 on LongBench (single QA Doc)**. The comparisons demonstrate that ProjScore consistently outperforms AttnScore across different settings of adopted techniques, isolating the novel contribution of AnDPro.

Our discoveries are as follows. (1) To achieve a better semantic succession, we merge a certain number of adjacent tokens as one chunk. We test different chunk sizes $C \in \{1, 2, 4, 8, 16\}$ on the three datasets from the Single-Doc. QA task. The experiments are conducted on Mistral-7b-Instruct-v0.2. Results are in Figure 10. As a hyperparameter directly related to contextual semantics, we found that when the budget is small, smaller chunks will lead to better performance, and vice versa. In the main experiments, we set the chunk size to $4$. (2) Following SnapKV and Ada-KV, we retain the first token by default. Results show that this leads to a very slight performance gain. (3) We follow Ada-KV to allocate budget across all heads within each layer according to their scores. The results show that this operator is critical to the overall performance gain. The integration of the

Table 8: **Ablation results of Qwen2-7B-Instruct on LongBench (single QA Doc)**. The comparisons demonstrate that ProjScore consistently outperforms AttnScore across different settings of adopted techniques, isolating the novel contribution of AnDPro.

| | Budget | Nrtvqa | Qasper | MF-en | Avg. | Nrtvqa | Qasper | MF-en | Avg. |
|---|---|---|---|---|---|---|---|---|---|
| | | CrossHead + Chunk + FirstToken + ProjScore | | | | CrossHead + Chunk + FirstToken +AttenScore | | | |
| A | 128 | 23.95 | 35.67 | 44.72 | **34.78** | 21.69 | 35.07 | 45.74 | 34.17 |
| | 256 | 22.52 | 39.90 | 46.20 | 36.21 | 24.25 | 39.16 | 46.87 | **36.76** |
| | 512 | 21.52 | 43.93 | 46.74 | 37.40 | 22.96 | 43.32 | 46.02 | **37.43** |
| | 1024 | 22.07 | 44.95 | 46.69 | 37.90 | 23.44 | 44.38 | 47.37 | **38.40** |
| | | Chunk + FirstToken +ProjScore | | | | Pooling + FirstToken + AttenScore | | | |
| B | 128 | 20.49 | 33.72 | 42.26 | **32.16** | 20.43 | 31.45 | 42.80 | 31.56 |
| | 256 | 21.25 | 37.88 | 43.43 | 34.19 | 22.28 | 36.70 | 43.95 | **34.31** |
| | 512 | 23.88 | 40.25 | 46.89 | **37.01** | 22.80 | 41.12 | 44.38 | 36.10 |
| | 1024 | 23.40 | 42.24 | 46.51 | 37.38 | 24.58 | 43.55 | 46.95 | **38.36** |
| | | CrossHead + FirstToken +ProjScore | | | | CrossHead + FirstToken +AttenScore | | | |
| C | 128 | 21.84 | 35.04 | 44.64 | **33.84** | 17.07 | 32.11 | 39.11 | 29.43 |
| | 256 | 22.30 | 38.72 | 44.31 | **35.11** | 23.36 | 36.89 | 43.24 | 34.50 |
| | 512 | 25.02 | 43.50 | 47.55 | **38.69** | 22.43 | 40.82 | 42.02 | 35.09 |
| | 1024 | 23.93 | 43.18 | 46.78 | **37.96** | 23.19 | 40.62 | 46.28 | 36.70 |
| | | CrossHead + Chunk +ProjScore | | | | CrossHead + Pooling +AttnScore | | | |
| D | 128 | 22.16 | 34.32 | 47.45 | **34.64** | 20.54 | 32.82 | 40.26 | 31.21 |
| | 256 | 25.12 | 39.42 | 45.01 | **36.52** | 22.86 | 35.62 | 43.93 | 34.14 |
| | 512 | 24.30 | 42.95 | 46.68 | **37.98** | 23.44 | 41.50 | 44.64 | 36.53 |
| | 1024 | 24.56 | 45.52 | 45.71 | **38.60** | 23.75 | 42.82 | 47.91 | 38.16 |
| | | ProjScore | | | | AttnScore | | | |
| E | 128 | 21.32 | 33.44 | 41.42 | **32.06** | 19.40 | 30.84 | 34.30 | 28.18 |
| | 256 | 21.67 | 36.45 | 45.76 | **34.63** | 21.34 | 31.67 | 38.96 | 30.66 |
| | 512 | 24.02 | 41.40 | 44.30 | **36.57** | 21.68 | 34.55 | 39.60 | 31.94 |
| | 1024 | 23.37 | 43.13 | 47.61 | **38.04** | 23.31 | 39.16 | 42.60 | 35.02 |

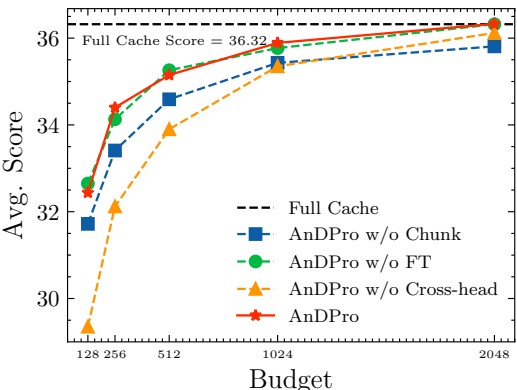

Figure 9: **Ablation Study on the adopted techniques.**

aforementioned techniques results in the high performance of AnDPro. Notably, AnDPro outperforms full cache with a budget size $k = 2048$.

Table 9: **Ablation results of Meta-Llama-3.1-8B-Instruct on LongBench (single QA Doc)**. The comparisons demonstrate that ProjScore consistently outperforms AttnScore across different settings of adopted techniques, isolating the novel contribution of AnDPro.

| | Budget | Nrtvqa | Qasper | MF-en | Avg. | Nrtvqa | Qasper | MF-en | Avg. |
|---|---|---|---|---|---|---|---|---|---|
| | | CrossHead + Chunk + FirstToken + ProjScore | | | | CrossHead + Chunk + FirstToken +AttenScore | | | |
| A | 128 | 26.66 | 38.22 | 54.54 | **39.81** | 27.19 | 38.01 | 53.13 | 39.44 |
| | 256 | 27.28 | 42.09 | 54.72 | **41.36** | 26.36 | 41.94 | 54.77 | 41.02 |
| | 512 | 28.03 | 44.25 | 55.61 | 42.63 | 29.12 | 44.86 | 54.86 | **42.95** |
| | 1024 | 29.40 | 46.83 | 56.72 | **44.32** | 28.14 | 46.77 | 55.81 | 43.57 |
| | | Chunk + FirstToken +ProjScore | | | | Pooling + FirstToken + AttenScore | | | |
| B | 128 | 24.87 | 32.33 | 52.51 | **36.57** | 25.54 | 31.40 | 50.99 | 35.98 |
| | 256 | 27.04 | 38.52 | 54.21 | **39.92** | 26.92 | 38.19 | 52.62 | 39.24 |
| | 512 | 27.24 | 42.18 | 54.25 | 41.22 | 28.40 | 41.52 | 55.40 | **41.77** |
| | 1024 | 28.57 | 45.80 | 54.89 | 43.09 | 28.32 | 45.37 | 56.81 | **43.50** |
| | | CrossHead + FirstToken +ProjScore | | | | CrossHead + FirstToken +AttenScore | | | |
| C | 128 | 28.51 | 38.05 | 52.92 | **39.83** | 26.78 | 34.67 | 52.32 | 37.92 |
| | 256 | 26.72 | 41.48 | 55.23 | **41.14** | 27.05 | 40.23 | 54.29 | 40.52 |
| | 512 | 29.79 | 43.50 | 55.11 | **42.80** | 28.96 | 43.11 | 54.61 | 42.23 |
| | 1024 | 31.48 | 45.54 | 56.75 | **44.59** | 29.98 | 46.06 | 55.32 | 43.79 |
| | | CrossHead + Chunk +ProjScore | | | | CrossHead + Pooling +AttenScore | | | |
| D | 128 | 30.13 | 37.46 | 54.01 | **40.53** | 25.06 | 34.50 | 52.27 | 37.28 |
| | 256 | 29.70 | 42.56 | 55.59 | **42.62** | 27.04 | 39.24 | 54.41 | 40.23 |
| | 512 | 31.08 | 44.48 | 55.62 | **43.73** | 26.92 | 42.68 | 55.20 | 41.60 |
| | 1024 | 31.11 | 46.05 | 56.39 | **44.52** | 28.52 | 46.01 | 55.72 | 43.42 |
| | | ProjScore | | | | AttnScore | | | |
| E | 128 | 25.19 | 32.36 | 51.41 | **36.32** | 28.10 | 28.56 | 42.69 | 33.12 |
| | 256 | 26.07 | 37.74 | 52.05 | **38.62** | 29.56 | 31.00 | 44.19 | 34.92 |
| | 512 | 28.51 | 41.88 | 55.28 | **41.89** | 28.91 | 36.00 | 51.55 | 38.82 |
| | 1024 | 27.75 | 45.54 | 56.22 | **43.17** | 29.51 | 40.70 | 54.31 | 41.51 |

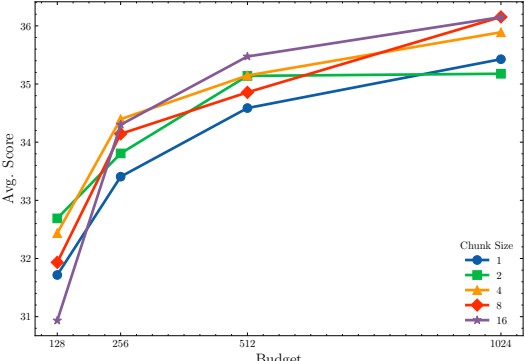

Figure 10: Comparison of experimental results with different chunk sizes on Single-Doc. QA

## C.6 Different Choices of $\theta$ and $b$

Recall that our projection-based scoring function is defined in the form $s_i = a_i(\boldsymbol{y}^\top \boldsymbol{v}_i + b)$. In this section, we investigate the effect of different choices of $\boldsymbol{\theta}$ and $b$. We conduct these experiments using Mistral-7B-Instruct-v2.0 on the datasets from the Single-Document QA task with the Mistral model.

To investigate the effect of the anchor direction $\boldsymbol{\theta}$, we parameterize the anchor direction as $\theta = \sum_{i=1}^{n} \tilde{a}_i v_i$ where $\tilde{a}_i = \text{Softmax}(q^T k_i/\alpha)$ with a temperature coefficient $\alpha$. This generalization covers several cases: $\alpha = 1$ corresponds exactly to our proposed anchor direction $\boldsymbol{\theta} = \boldsymbol{y}$ ; $\alpha \to \infty$ corresponds to the average of all value vectors; an $\alpha \to 0$ corresponds to the value vector with the highest attention weight. The results are in Table 10, demonstrating that setting the anchor direction as $\boldsymbol{\theta} = \boldsymbol{y}$ achieves a robust performance.

Table 10: Results of different choices of the anchor direction $\boldsymbol{\theta} = \sum_{i=1}^{n} \tilde{a}_i \boldsymbol{v}_i$, where $\tilde{a}_i = \text{Softmax}(\boldsymbol{q}^\top \boldsymbol{k}_i/\alpha)$ with a temperature coefficient $\alpha$. The used model is Mistral-7B-Instruct-v0.2 and the dataset is LongBench (single QA Doc). The results demonstrate that setting the anchor direction as $\boldsymbol{\theta} = \boldsymbol{y}$ achieves a robust performance.

| | Nrtvqa | Qasper | MF-en | Avg. | Nrtvqa | Qasper | MF-en | Avg. |
|---|---|---|---|---|---|---|---|---|
| | Budget=128 | | | | Budget=256 | | | |
| $\alpha = 0$ (highest attention) | 15.79 | 14.67 | 26.22 | 18.89 | 16.01 | 15.05 | 28.52 | 19.86 |
| $\alpha = 0.5$ | 23.10 | **25.68** | 47.69 | 32.16 | 23.15 | 27.12 | 48.82 | 33.03 |
| $\alpha = 1$ ($\boldsymbol{\theta} = \boldsymbol{y}$) | **24.84** | 25.14 | 47.33 | **32.44** | **25.71** | **28.17** | **49.31** | **34.40** |
| $\alpha = 1.5$ | 22.81 | 23.47 | **47.99** | 31.42 | 24.62 | 28.08 | 48.35 | 33.68 |
| $\alpha = \infty$ (average) | 15.83 | 15.75 | 27.96 | 19.85 | 16.62 | 16.06 | 28.96 | 20.55 |
| | Budget=512 | | | | Budget=1024 | | | |
| $\alpha = 0$ (highest attention) | 16.24 | 14.56 | 29.83 | 20.21 | 16.96 | 15.57 | 34.68 | 22.40 |
| $\alpha = 0.5$ | 24.87 | 29.80 | 49.33 | 34.67 | **25.89** | 30.57 | 48.93 | 35.13 |
| $\alpha = 1$ ($\boldsymbol{\theta} = \boldsymbol{y}$) | 24.87 | **30.99** | **49.58** | **35.15** | 25.87 | **32.54** | 49.35 | **35.92** |
| $\alpha = 1.5$ | **25.13** | 30.34 | 48.14 | 34.54 | 25.81 | 31.57 | **49.64** | 35.67 |
| $\alpha = \infty$ (average) | 16.81 | 17.62 | 33.50 | 22.64 | 17.11 | 21.56 | 38.68 | 25.78 |

We then investigate the impact of $b$, and the results are in Figure 11. We find that though a careful choice of $b$ promise lead to a better results, simply setting $b = 0$ is enough to derive a stable performance. Therefore, we set $b = 0$ in our main experiments, without the need for labor-intensive hyperparameter tuning. Moreover, when $b$ approaches infinity (e.g., $b = 1000$), the scoring function converges to attention-based ones, and the performance approaches the performance of Ada-KV.

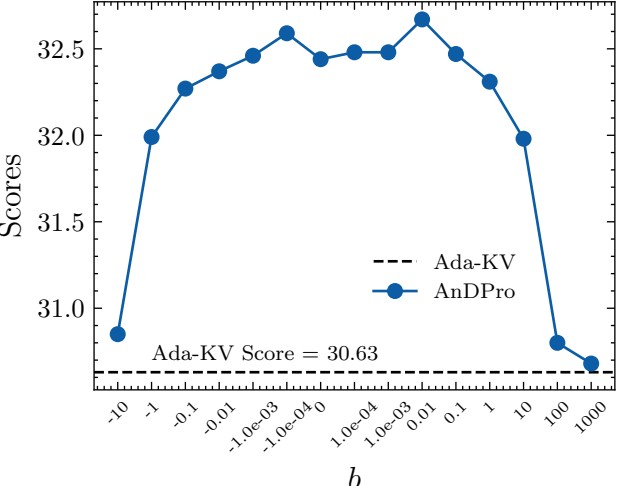

Figure 11: Different choices of $b$

## C.7  Eviction Losses and Cosine Similarity

Figure 12 illustrates the eviction loss and cosine similarity derived from 2 methods under different budget allocations, with the eviction loss quantified as $\frac{||y-\hat{y}||^2}{||y||^2}$. The results show that AnDPro consistently yields a lower eviction loss, and a larger cosine similarity, which can to some extend explain the high performance of AnDPro.

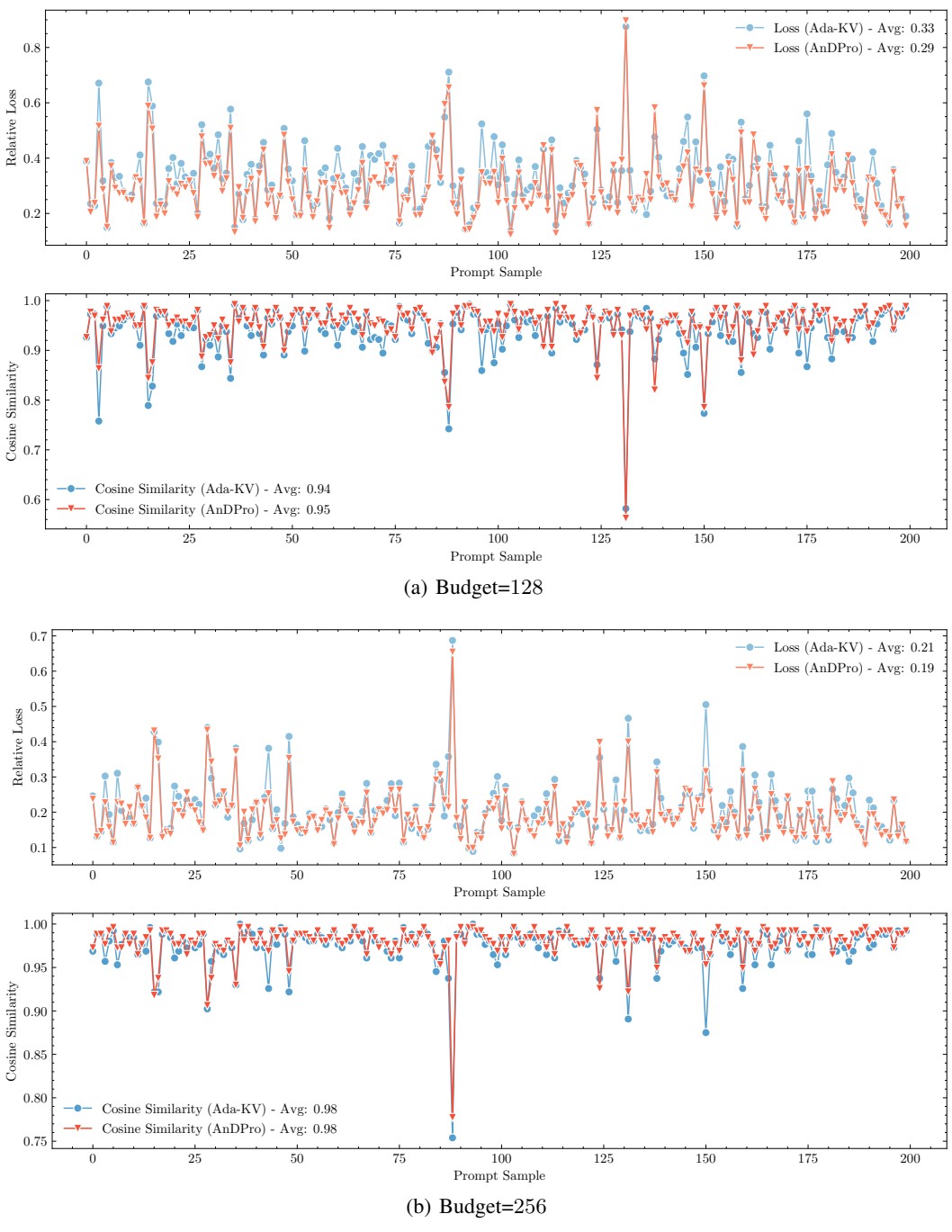

(a) Budget=128

(b) Budget=256

Figure 12: **Comparison of eviction loss and cosine similarity**. Experiments are conducted using Mistral7B-instruct-v0.2 on 200 samples from the Qasper dataset.

## C.8 Token Visualization

Figure 13 illustrates the distribution of our defined scoring functions across various layers (using Mistral7B-instruct-v0.2 on a sample from the Qasper Dataset), demonstrating that the distribution exhibits a sparsity feature like that of attention weights.

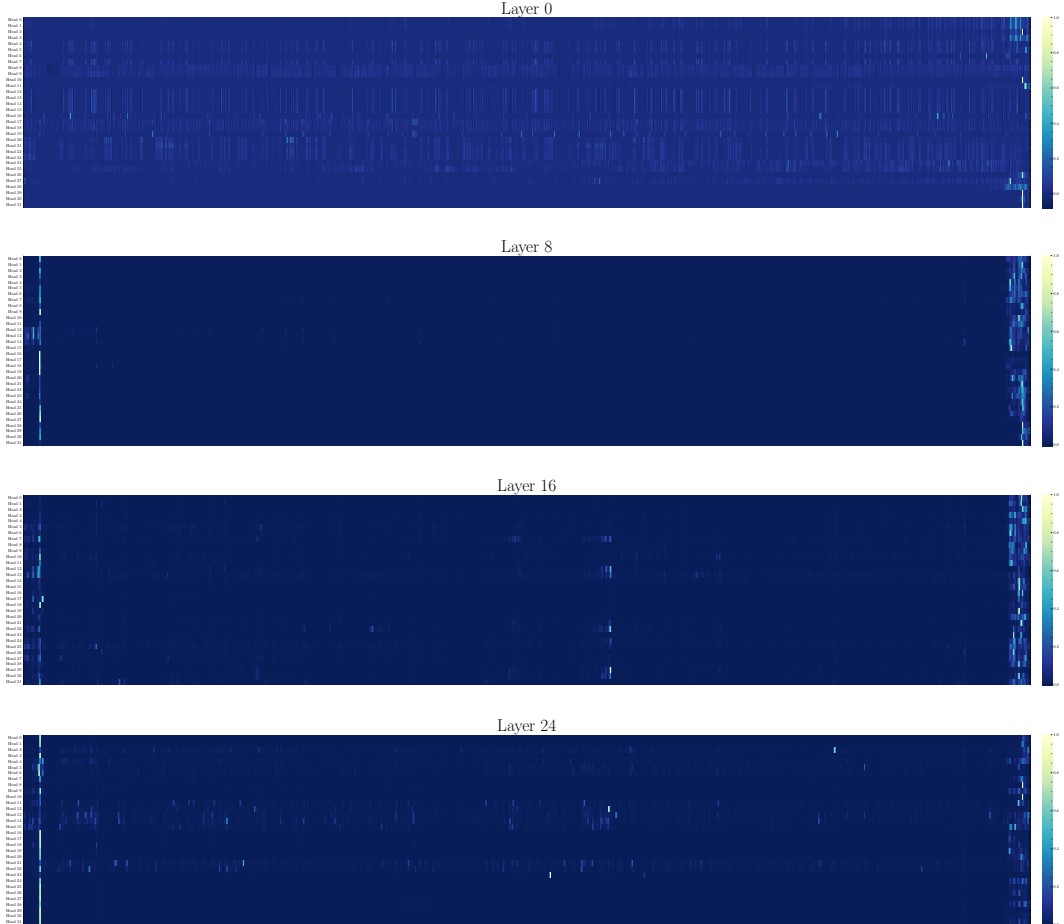

Figure 13: Visualization of Token Scores for Different Layers.

## C.9 Value Vectors Visualization & Magnitude Histogram

We conduct two experiments to intuitively explain why projection-based methods outperform attention-based approaches. First, we provide the PCA visualization of value vectors in Figure 14. It reveals that projection methods preserve geometrically critical tokens along the original output direction, minimizing post-eviction semantic drift. In contrast, attention-based selection disrupts the latent spatial distribution, causing divergent outputs. Second, complementary analysis in Figure 15 demonstrates significant variance of the vector vector magnitude. This indicates that the value vectors contain rich semantic information, which is ignored in attention-based approaches. This further underscores the necessity of projection-based strategies.

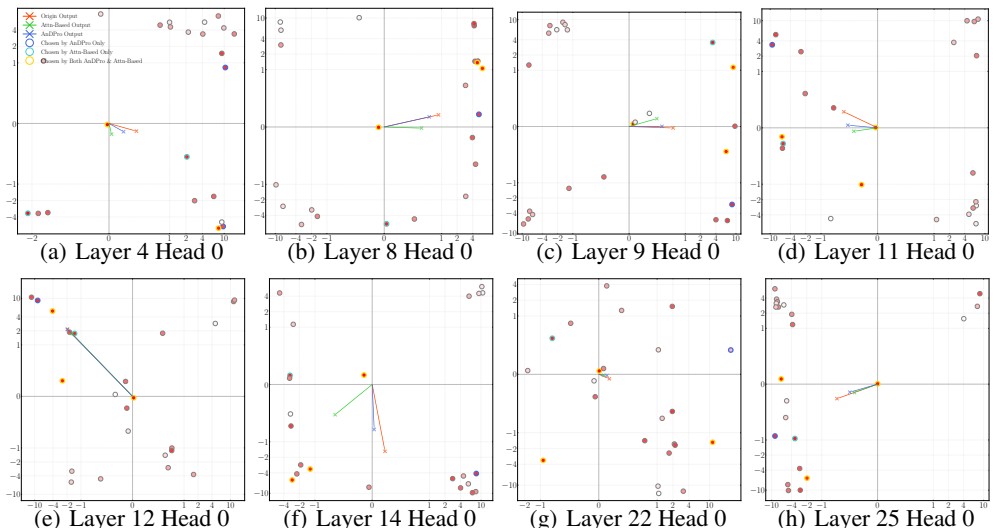

(a) Layer 4 Head 0    (b) Layer 8 Head 0    (c) Layer 9 Head 0    (d) Layer 11 Head 0

(e) Layer 12 Head 0    (f) Layer 14 Head 0    (g) Layer 22 Head 0    (h) Layer 25 Head 0

Figure 14: The visualization of token value vectors and outputs of different approaches using PCA dimensionality reduction. We use an easy case with 23 tokens. Each dot represents a token value vector, with color intensity (red hues) indicating attention scores. Tokens selected by different methods are highlighted by colored outer rings. Arrow vectors depict output directions. We visualize the results from the Head 0 across several layers. The results demonstrate that the projection-based approach can achieve smaller distances between the pre- can post-eviction outputs by maintaining the semantical directions of the original outputs.

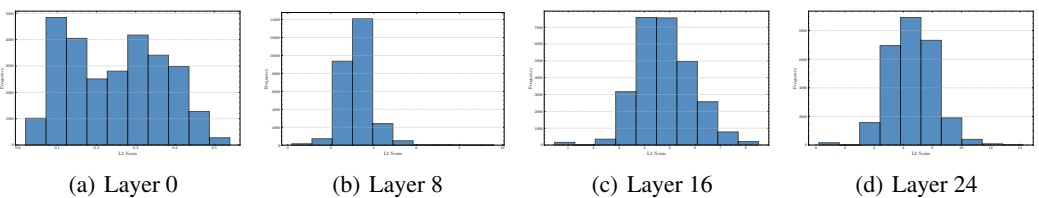

(a) Layer 0      (b) Layer 8      (c) Layer 16      (d) Layer 24

Figure 15: The distribution of the value vector magnitudes from several different layers in a test example. This reveals the significant variability and underscores the important semantic information captured by projection but overlooked by pure attention methods.

## C.10    Case Study

We conduct case study on some simple test cases. We calculate the total scores of each token in the prefilling Phase based on the AnDPro and attention weights. Some results are presented in Figure 16. These token-level scores are scaled to the range of 0 to 1 and sorted in descending order. Specifically, a higher score indicates that the token is considered more important.

By observing the key tokens marked in red (identified by humans), AnDPro places better emphasis on the key tokens compared to the attention-based approach, thereby intuitively demonstrating the effectiveness of our method.

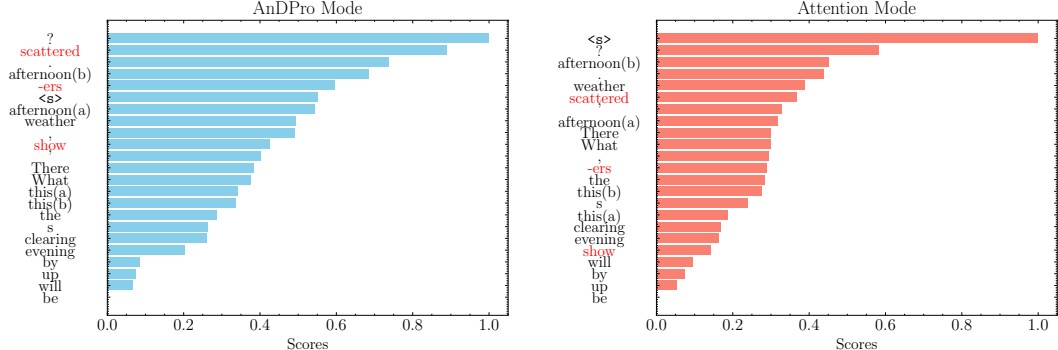

**Prompt 1:** There will be scattered showers this afternoon, clearing up by evening. What's the weather this afternoon?

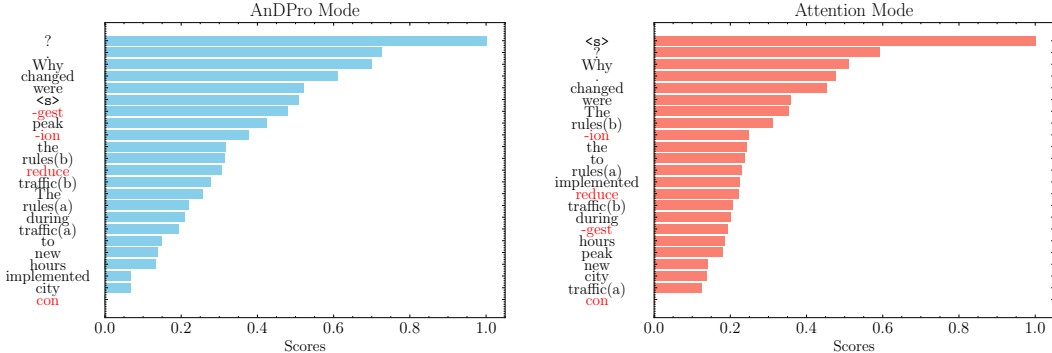

**Prompt 2:** The city implemented new traffic rules to reduce congestion during peak hours. Why were the traffic rules changed?

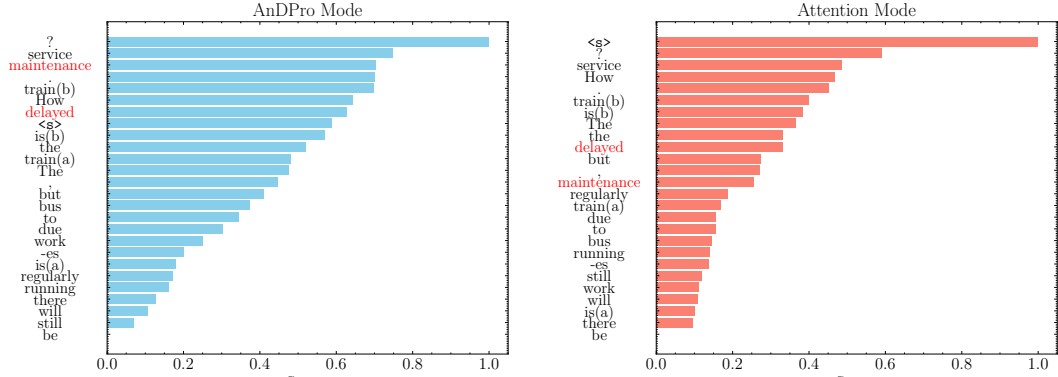

**Prompt 3:** The train is delayed due to maintenance work, but there will still be buses running regularly. How is the train service?

Figure 16: **Case study.** We visualize the importance orders given by different approaches, AnDPro, and attention-based approach. Red tokens are key tokens identified by humans.

## C.11 Long Context

We have conducted additional experiments on very long sequences using the Needle-in-a-Haystack benchmark, extending the test sequence length from the original 32K tokens up to 384K tokens. Specifically, we use the Llama-3.1-8B-Instruct model (pretrain context length 128K) under a budget size of 128. Results are in Figure 17.

The results demonstrate that AnDPro consistently achieves superior performance even on long sequences. Notably, when sequence length exceeds 256K tokens, Full cache fails due to memory constraints, while AnDPro remains effective. Although all methods fail beyond 512K tokens, AnDPro maintains top performance within feasible length ranges.

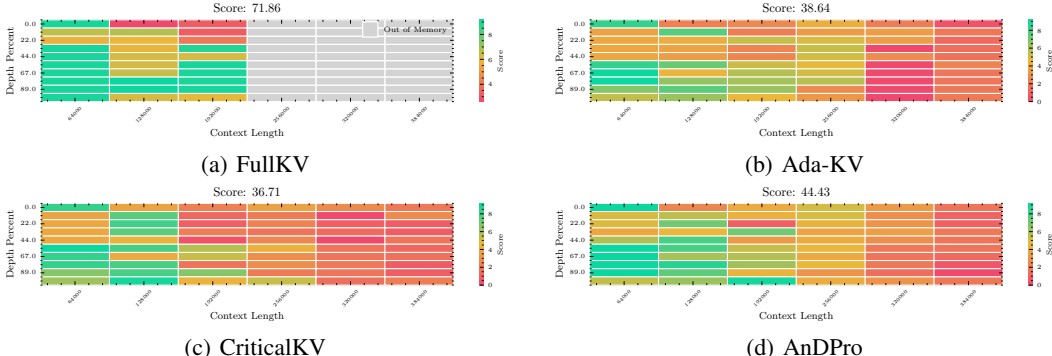

Figure 17: Results of Llama-3.1-8B-instruct on the Needle-in-a-Haystack test with extremely long sequences (up to 384K). The x-axis represents the document's context length, and the y-axis indicates the depth of the needle insertion. Each square in the heatmap demonstrates the correctness of retrieving a proper sentence inserted in the context for a given context length and insertion depth. The Average Score is calculated by averaging the sum of score with different insert depth across various context lengths. Gray squares indicate out-of-memory errors due to excessive context length.

## C.12 Large-Sized LLMs

To evaluate the generalization and applicability of AnDPro on larger models, we conduct additional experiments using the Qwen2.5-32B-Instruct model on the LongBench single-document QA benchmark. The results are in Table 11.

Table 11: Results of Qwen2.5-32B-Instruct on the LongBench single-document QA benchmark.

|         | Budget= 128 | Budget= 256 | Budget= 512 | Budget= 1024 |
|---------|-------------|-------------|-------------|--------------|
| Ada-KV  | 34.66       | 38.02       | 41.57       | 43.82        |
| AnDPro  | 38.63       | 40.91       | 43.62       | 45.81        |

## C.13 Long Decoding Tasks

To further demonstrate the broader applicability of our method, we conduct additional experiments on long decoding tasks, particularly in LLM reasoning settings. For a better comparison with the baselines, we additionally implemented the decoding versions of AnDPro and SnapKV, while Ada-KV does not have a decoding version. Specifically, we evaluated our approach on the AIME24 dataset using DeepSeek-R1-Distill-Qwen-14B.

Table 12 reports the reasoning performance (accuracy, %) under different KV cache budget settings. Table 13 reports the throughput (tokens/s) and maximum supported batch size (OOM threshold) during 10K-token generation.

Table 12: Performance (accuracy, %) of different methods on the AIME24 dataset under two KV cache budget settings (2K and 4K).

| | Budget= $2K$ | Budget= $4K$ |
|---|---|---|
| SnapKV | 50.00 | 53.33 |
| AnDPro | 60.00 | 63.33 |

Table 13: Throughput (tokens/s) and maximum supported batch size (before out-of-memory occurs) for different methods during 10K-token generation on a single GPU with 80GB of VRAM.

| | Generation Length | Max Batch Size (OOM Threshold) | Throughput (token/s) |
|---|---|---|---|
| Full | 10K | 28 | 181.09 |
| SnapKV | 10K | 80 | 415.48 |
| AnDPro | 10K | 80 | 413.15 |

# D   Limitation and Broader Impact

**Limitation**    We have limited this work to the setting of uniform budget allocation across layers. It remains to be investigated how to perform fine-grained budget allocation between layers, building upon existing studies that have demonstrated the varying importance of different layers.

**Broader Impact**    Our methodology pioneers a novel paradigm for KV Cache Compression by incorporating directional characteristics of value vectors (beyond attention weight exclusivity), which we believe will catalyze transformative breakthroughs in this field. Crucially, the proposed technique achieves substantial inference acceleration with only marginal performance degradation, establishing its deployment potential for edge computing scenarios and long-context sequence processing.

