# OpenReview forum: "Accurate KV Cache Eviction via Anchor Direction Projection for Efficient LLM Inference"
_NeurIPS.cc/2025/Conference — NeurIPS 2025 poster_

### Official Review · Reviewer_AuWV · 2025-06-11

**Clarity:** 3
**Significance:** 3
**Originality:** 4
**Rating:** 5
**Confidence:** 4

**Summary:**

This paper proposes AnDPro (Anchor Direction Projection), a novel KV cache eviction strategy for large language models that goes beyond traditional attention-weight-based heuristics by incorporating geometric information from value vectors. It defines token importance via a projection-based scoring function, $\( s_i = a_i \cdot y^\top v_i \)$, where $\( y \)$ is the pre-eviction output direction. This approach is derived from a sparse optimization formulation and efficiently implemented with support for cross-head cache budgeting. Experiments on LongBench and Needle-in-a-Haystack benchmarks show that AnDPro outperforms prior methods, achieving 96.07% of full-cache accuracy using only 3.44% of the cache, reducing memory by up to 46% without quality loss.

**Questions:**

1. How is y computed in Section 3.4? Is it computed with KV cache in the local window and what is the window size? Do you compute y before each step of generation?
2. Do different attention heads select the same tokens to keep? Or they can choose to preserve different tokens?
3. Will the evicted token be permanently deleted from the KV cache? Or they can still be selected and attended to at future generation steps?
4. What is the chunk size in Section 3.4?

**Ethical Concerns:**

["NO or VERY MINOR ethics concerns only"]

**Final Justification:**

My questions have been well resolved by the authors' rebuttal. And I will keep my original recommendation.

**Limitations:**

yes

**Paper Formatting Concerns:**

N/A.

**Quality:**

4

**Strengths And Weaknesses:**

[Strengths]
1. The paper proposes a novel projection-based token importance score  $\( s_i = a_i \cdot y^\top v_i \)$​, incorporating value vector geometry beyond attention weights.
2. The paper provides comprehensive theoretical analysis/proof and intuitive explanation for the proposed method.
3. The authors evaluate the proposed method across different accuracy benchmarks with a wide range of LLMs, demonstrating the effectiveness of their optimizations, achieving 96.07% of full-cache accuracy using only 3.44% of the cache, reducing memory by up to 46% without quality loss.


[Weaknesses]
1. The discussion of implementation details should be improved.

---

> ### Author Rebuttal · Authors · 2025-07-31
>
> Dear Reviewer AuWV,
>
> Thank you for your insightful comments. We sincerely hope that our rebuttal could properly address your concerns. If so, we would deeply appreciate it if you consider raising your score.
>
> **Weakness 1：The discussion of implementation details should be improved.**
>
> We respond to the questions regarding implementation details as follows, and we will provide a more detailed description in the revised manuscript. Please let us know if you have any further questions.
>
> **Question 1. How is $y$ computed?**
>
> > How is $y$ computed in Section 3.4? Is it computed with KV cache in the local window and what is the window size? Do you compute $y$ before each step of generation?
>
> We clarify the calculation process as follows:
>
> - **Definition of $y$:** The vector y is an attention output calculated as the **weighted sum** of all **value vectors ($v_i$)**, using the corresponding **attention scores ($a_i$) as weights**.
> - **Use of Local Window:** In the local window computation, we calculate the vector y by performing an attention operation between the **queries**  from the **window** and the **full KV cache** that covers all historical information. The local observation window size used in our experiments is **32**.
> - **Timing of Computation:** Our method **computes and applies y only once during the prefilling stage**, i.e., immediately after the context is filled. During the subsequent **decoding** phase, the model directly uses the **pre-selected important tokens** without recomputing $y$ at each generation step.
>
> Since y is derived exclusively from the **uncompressed KV cache** within a fixed local window during prefilling, and remains fixed during decoding, its computation does not incur additional per-step cost in the generation process.
>
> **Question 2: Do different attention heads select the same tokens to keep**
>
> > Do different attention heads select the same tokens to keep? Or they can choose to preserve different tokens?
>
> Thank you for raising this detailed question.
>
>  During the pre-filling stage, each attention head **independently** selects important tokens based on its own computed attention scores. We employ an anchor projection method to process all heads in parallel, allowing each head to adaptively choose **different numbers and positions of tokens** according to its specific function and attention patterns.
>
> This design enables each head to fully leverage its distinct role rather than being constrained by a unified selection strategy, thereby enhancing the model’s ability to capture critical information and improving overall reasoning efficiency.
>
> **Question 3.**
>
> > Will the evicted token be permanently deleted from the KV cache? Or they can still be selected and attended to at future generation steps?
>
> - **Permanent Eviction**
>   -  To maintain a minimal memory footprint, we exclusively retain the **Key-Value** pairs for the tokens that are selected. This means that any token evicted from the **KV cache is permanently discarded** and cannot be accessed or attended to in any subsequent generation steps.
>   -  Our experimental results validate this strategy, demonstrating that by keeping only about **3.44%** of the original KV cache, we can still preserve approximately **96%** of the model's generation performance. This highlights our method's ability to achieve significant cache compression with negligible impact on quality.
>
> > What is the chunk size in Section 3.4?
>
> - **About Chunk Size**
>   -  The primary motivation for our chunking operation is to preserve **local semantic continuity**. By grouping adjacent tokens, which are typically semantically related, we ensure that our selection mechanism evaluates local contextual information as **a coherent unit** rather than as isolated tokens. It is also worth noting that because our method incorporates information from the **value states**, merely applying **pooling techniques** to the attention scores (as in methods like **SnapKV**) is insufficient for effectively enhancing the continuity of the selected tokens. **To address this, we designed a chunk operation that combines attention and value inner products, specifically by summing the weighted vectors corresponding to tokens within a chunk.** This approach effectively overcomes the aforementioned limitation.
>   -  In our experiments, we set the chunk **size to 4**. **Appendix C.5.2** presents the impact of different chunk sizes on the experimental results.

---

> > ### Comment · Reviewer_AuWV · 2025-08-04
> >
> > Thank you for the response. My questions have been well resolved by the authors' rebuttal. And I will keep my original recommendation.

---

> > > ### Author Response · Authors · 2025-08-04
> > > **Thank you for your kind support.**
> > >
> > > Dear Reviewer AuWV,
> > >
> > > Thanks for your kind support and for helping us improve the paper. We sincerely appreciate your valuable suggestions.
> > >
> > > Best regards,
> > >
> > > Authors

---

### Official Review · Reviewer_qyz2 · 2025-06-30

**Clarity:** 2
**Significance:** 3
**Originality:** 3
**Rating:** 4
**Confidence:** 3

**Summary:**

This paper proposes AnDPro, a novel KV cache eviction method for efficient LLM inference. Unlike prior approaches that rely solely on attention weights to measure token importance, AnDPro leverages the geometric relationships between token value vectors in the embedding space. It introduces a projection-based scoring function, where token importance is quantified by projecting value vectors onto an anchor direction. Evaluated on 16 LongBench datasets, AnDPro achieves 96.07% of full-cache accuracy using only 3.44% of the KV cache budget.

**Questions:**

Q1: What would AnDPro perform on larger-sized LLMs (e.g., 32B, 70B)?
Q2: What is the specific computational overhead of AnDPro? How does AnDPro’s overhead compare to the overhead of existing methods?

**Ethical Concerns:**

["NO or VERY MINOR ethics concerns only"]

**Final Justification:**

Thanks for the author's rebuttal, and I maintain the original positive rating.

**Limitations:**

Yes

**Quality:**

2

**Strengths And Weaknesses:**

S1: The theoretical analysis is novel and solid.

S2: The experimental comparison is comprehensive.

S3: The writing of this paper is easy to follow.

W1: The evaluation is only conducted on relatively small-sized LLMs.

W2: The specific computational overhead of AnDPro is not provided.

---

> ### Author Rebuttal · Authors · 2025-07-31
>
> Dear Reviewer qyz2,
>
> Thank you for your insightful comments. We sincerely hope that our rebuttal could properly address your concerns. If so, we would deeply appreciate it if you consider raising your score.
>
> **W1 & Q1: Large-sized LLMs.**
>
> > W1: The evaluation is only conducted on relatively small-sized LLMs.
> >
> > Q1: What would AnDPro perform on larger-sized LLMs (e.g., 32B, 70B)?
>
> Thank you for your interest in the scalability of our approach. To evaluate the **generalization and applicability** of AnDPro on larger models, we conducted additional experiments using the **Qwen2.5-32B-Instruct** model on the **LongBench single-document QA** benchmark.
>
> | Method | Budget=128 | Budget=256 | Budget=512 | Budget=1024 |
> | ------ | ---------- | ---------- | ---------- | ----------- |
> | Ada-KV | 34.66      | 38.02      | 41.57      | 43.82       |
> | AnDPro | 38.63      | 40.91      | 43.62      | 45.81       |
>
> The results in the table above clearly indicate that across all tested average per-layer per-head computational budgets (from 128 to 1024), **AnDPro** consistently outperforms **Ada-KV**. This demonstrates that our method successfully generalizes to **larger-scale models** while maintaining its superior performance.
>
> We will include the **detailed results and analysis** in the final version of the paper.
>
> **W2 & Q2. Computational overhead.**
>
> > W2: The specific computational overhead of AnDPro is not provided.
> >
> > Q2: What is the specific computational overhead of AnDPro? How does AnDPro’s overhead compare to the overhead of existing methods?
>
> Thank you for your detailed question. To ensure decoding efficiency, our method does **not** perform eviction at every decoding step. Instead, **the update is executed only once during the prefilling stage**, which significantly reduces runtime overhead.
>
> As analyzed in **Figure 7 (Appendix C.4)**, we evaluated the runtime overhead of our method. To provide a more detailed quantitative breakdown, the specific runtime measurements are presented below (unit: seconds/milliseconds):
>
> | Input Length 64K | Prefilling (s) | KV update (s) | Layer-wise KV update (ms) |
> | ---------------- | -------------- | ------------- | ------------------------- |
> | Full             | 9.45           | --            | --                        |
> | SnapKV           | 9.50           | 0.10          | 3.05                      |
> | Ada-KV            | 9.88           | 0.43          | 13.28                     |
> | AnDPro           | 10.04          | 0.58          | 18.13                     |
>
> | Input Length 128K | Prefilling (s) | KV update (s) | Layer-wise KV update (ms) |
> | ----------------- | -------------- | ------------- | ------------------------- |
> | Full              | 28.78          | --            | --                        |
> | SnapKV            | 29.00          | 0.11          | 3.42                      |
> | Ada-KV             | 29.47          | 0.59          | 18.31                     |
> | AnDPro            | 29.78          | 0.88          | 27.41                     |
>
> The results show that while **AnDPro's KV update stage** takes slightly longer than **SnapKV** and **Ada-KV**, this overhead is **negligible** when compared to the total **Prefilling** time. Therefore, the results quantitatively confirm that the additional computation from our method is **minimal and does not impact the overall decoding efficiency**.

---

> > ### Comment · Reviewer_qyz2 · 2025-08-05
> >
> > Thank you for your response, and I will keep my positive recommendation.

---

> > > ### Author Response · Authors · 2025-08-05
> > >
> > > Dear Reviewer qyz2,
> > >
> > > Thanks for your kind support and for helping us improve the paper. We sincerely appreciate your valuable suggestions.
> > >
> > > Best regards,
> > >
> > > Authors

---

### Official Review · Reviewer_PCLe · 2025-07-02

**Clarity:** 3
**Significance:** 3
**Originality:** 3
**Rating:** 4
**Confidence:** 4

**Summary:**

This paper proposes AnDPro (Anchor Direction Projection), a new method for KV cache eviction in large language models (LLMs). Traditional methods often rely on attention weights to determine token importance, but these heuristics ignore the geometric structure of value vectors, potentially leading to suboptimal eviction decisions. AnDPro introduces a projection-based scoring function that measures the importance of each token by projecting its value vector onto an “anchor direction”—the direction of the pre-eviction output. The paper formulates the eviction process as an integer optimization problem and derives a relaxed convex formulation to inspire the design of the scoring function. Experimental results on two models (Mistral-7B and LLaMA3-8B) and across 16 tasks in LongBench and the Needle-in-a-Haystack benchmark demonstrate that AnDPro achieves superior accuracy and efficiency, maintaining over 96% of full-cache performance using only 3.44% of the cache.

**Questions:**

1. Can this method be extended to decoding phase? As it mainly focuses on the prefilling stage.
2. What is the computational overhead of computing projections and scores per token? While the authors claim negligible latency, quantitative timing comparisons of the Update KV phase across methods would support the claim.

**Ethical Concerns:**

["NO or VERY MINOR ethics concerns only"]

**Limitations:**

Yes

**Quality:**

3

**Strengths And Weaknesses:**

Strength:
1. The authors model KV cache eviction as a constrained combinatorial optimization problem (P0-C), then relax it to a sparse convex problem (P1). This provides a mathematically grounded approach to derive a new scoring function based on the vector space geometry of the value states, rather than relying on attention weights alone.
2. The authors evaluate AnDPro on two strong LLMs across diverse tasks and memory budgets, demonstrating consistent improvement over prior methods including SnapKV, Pyramid, Ada-KV, and CriticalKV. Notably, it achieves SOTA performance even at low memory budgets.
3. They also provided a comprehensive analysis in the appendix.

Weakness:
1. The gap between the original non-convex objective (P0-C) and the relaxed convex form (P1) is not analyzed. Specifically, there is no theoretical or empirical discussion of how close the solution to P1 is to the true solution of P0-C.
2. The reference of some methods, e.g. PyramidKV are missing.

---

> ### Author Rebuttal · Authors · 2025-07-31
>
> Dear Reviewer PCLe,
>
> Thank you for your insightful comments. We sincerely hope that our rebuttal could properly address your concerns. If so, we would deeply appreciate it if you consider raising your score.
>
> **Weakness 1. Theoretical analysis.**
>
> > The gap between the original non-convex objective (P0-C) and the relaxed convex form (P1) is not analyzed. Specifically, there is no theoretical or empirical discussion of how close the solution to P1 is to the true solution of P0-C.
>
> - Our analysis follows a structured derivation path:
>   1. The transformation from **(P0-C) to (P0-R)** is a standard technique in traditional convex optimization, and when the penalty parameter $\lambda$ is sufficiently large, the two formulations are **equivalent** in terms of support recovery.
>   2. The reformulation from **(P0-R) to Eq. (9)** is an **exact transformation**.
>   3. The relaxation from **Eq. (9) to (P1)** is a common and principled convex relaxation strategy that preserves the sparse structure and makes further analysis possible.
> - While the final relaxation (P1) does not always yield an exact solution to (P0-C), such a **relaxation is necessary** due to the **NP-hardness** of the original problem. Performing a precise theoretical gap analysis between (P1) and (P0-C) is thus **nontrivial** and remains an open challenge.
> - Importantly, our use of this relaxation serves as a **theoretical foundation** for deriving a **heuristic scoring function**, which is the core of our projection-based method. While we do not directly solve (P1), experiments show that the resulting heuristic performs **strongly in practice**, with low eviction loss and robust empirical results (Appendix C.7).
>
> **Weakness 2. Miss reference.**
>
> > The reference of some methods, e.g. PyramidKV are missing.
>
> Thank you for pointing out the missing references. We will add the missing references, such as **PyramidKV**, to the **Related Work** section and ensure they are cited at their first mention in the revised manuscript.
>
> **Question 1. Extension to decoding phase.**
>
> > Can this method be extended to decoding phase? As it mainly focuses on the prefilling stage.
>
> Thank you for your valuable question. While our method was originally designed for the **prefilling** stage, we have **explored its extension to the decoding phase**. For a better comparison with the baselines, we additionally implemented the decoding versions of AnDPro and SnapKV, while Ada-KV does not have a decoding version.
>
> We conduct experiments on the AIME24 dataset using the **DeepSeek-R1-Distill-Qwen-14B**. We report the  performance in **Table 1**.
>
> Table 1: Performance (accuracy, %) of different methods on the AIME24 dataset under two KV cache budget settings (2K and 4K). The performance of the **full cache** method is **66.66**. Higher values indicate better reasoning performance. For a budget of 2k, the **KV cache update interval** is 1k, and for a budget of 4k, the **update interval** is 2k.
>
> | **Method** | **Budget=2k** | **Budget=4k** |
> | ---------- | ------------- | ------------- |
> | Snapkv     | 50.00         | 53.33         |
> | Andpro     | 60.00         | 63.33         |
>
> The results show that **AnDPro maintains high inference accuracy** despite the additional computations introduced during decoding.
>
> **Question 2. Computational overhead.**
>
> > What is the computational overhead of computing projections and scores per token? While the authors claim negligible latency, quantitative timing comparisons of the Update KV phase across methods would support the claim.
>
> Thank you for raising the question regarding the computational overhead of per-token projection and scoring. To ensure decoding efficiency, our method does **not** perform eviction at every decoding step. Instead, **the update is executed only once during the prefilling stage**, which significantly reduces runtime overhead.
>
> As analyzed in **Figure 7 (Appendix C.4)**, we measured the additional time cost introduced by our approach. The measured runtime values are as follows (unit: seconds/milliseconds):
>
> Table 2: Runtime analysis of different methods during prefilling on sequences of length **64K** and **128K**. Although AnDPro introduces slightly higher update latency compared to SnapKV and AdaKV, the additional overhead is **negligible relative to the total prefilling time**.
>
> | 64K    | Prefilling (s) | KV update (s) | Layer-wise KV update (ms) |
> | ------ | -------------- | ------------- | ------------------------- |
> | Full   | 9.45           | --            | --                        |
> | SnapKV | 9.50           | 0.10          | 3.05                      |
> | AdaKV  | 9.88           | 0.43          | 13.28                     |
> | AnDPro | 10.04          | 0.58          | 18.13                     |
>
> | 128K   | Prefilling (s) | KV update (s) | Layer-wise KV update (ms) |
> | ------ | -------------- | ------------- | ------------------------- |
> | Full   | 28.78          | --            | --                        |
> | SnapKV | 29.00          | 0.11          | 3.42                      |
> | AdaKV  | 29.47          | 0.59          | 18.31                     |
> | AnDPro | 29.78          | 0.88          | 27.41                     |
>
> The results confirm that the additional computation introduced by the anchor direction update is **minimal and does not impact the overall decoding efficiency**. These findings are also discussed in the revised manuscript, where we clarify that our method maintains runtime performance comparable to existing baselines.

---

### Official Review · Reviewer_iGUk · 2025-07-02

**Clarity:** 3
**Significance:** 3
**Originality:** 4
**Rating:** 4
**Confidence:** 4

**Summary:**

The paper introduces AnDPro (Anchor Direction Projection), a novel method for improving Key-Value (KV) cache eviction in Large Language Models (LLMs). By proposing a projection-based scoring function, AnDPro aims to accurately measure token importance in the context of KV cache management, thus optimizing memory usage and inference latency. The authors argue that traditional methods relying on attention weights overlook spatial relationships between token value states in vector space, leading to suboptimal token selections. Through extensive experiments on 16 datasets from the LongBench benchmark, AnDPro demonstrates its effectiveness by maintaining high accuracy with significantly reduced KV cache budgets compared to state-of-the-art techniques.

**Questions:**

See weaknesses.

**Ethical Concerns:**

["NO or VERY MINOR ethics concerns only"]

**Final Justification:**

My recommendation is between borderline accept and accept.

The authors provided a strong rebuttal that resolved most of my key concerns.

Resolved: The new data on value vector non-orthogonality and the clarifications on computational overhead were convincing. The expanded evaluation on new models and a reasoning task has significantly strengthened the paper. I weigh these resolutions heavily.

Remaining: The main weakness, now minor, is that the evaluation could still be more comprehensive. While much improved, results across a wider variety of benchmarks would make the claims more definitive.

**Limitations:**

See weaknesses.

**Paper Formatting Concerns:**

No formatting concerns.

**Quality:**

4

**Strengths And Weaknesses:**

### Strengths:
1) AnDPro introduces an anchor direction projection technique that considers the spatial relationships within vector spaces, providing a more nuanced way to evaluate token importance beyond simple heuristics like attention weights, which is Innovative.
2) The paper offers a thorough theoretical analysis, including derivations and propositions, which underpin the design of the proposed scoring function.

### Weaknesses:
1) The idea of considering value states in the vector space is excellent. However, given that random vectors in high-dimensional spaces are almost orthogonal, and the vector space of value states is a very high-dimensional space, have you observed whether the value state vectors are orthogonal? If they are nearly orthogonal, then the illustration in Figure 1 could be misleading because the anchor direction would almost be dominated by attention weights, thereby reducing the accuracy gains obtained by the proposed method. If they are not orthogonal, please provide experimental results.
2) In the implementation, is the anchor direction updated during each forward pass of the LLM, and what is the associated computational overhead? Clarification is needed on whether the anchor direction is recalculated or updated during each forward pass of the LLM. The computational cost associated with this update should also be quantified to understand its impact on overall efficiency.
3) Could the compressed KV cache mislead the anchor direction compared to its uncompressed state? If this is the case, then the practical effectiveness of AnDPro might deviate from the theoretical derivation presented in Section 3.
4) Experiments were only conducted on Mistral-7B-Instruct-v0.2, which is insufficient to demonstrate that the method can generalize to other series of models or models of different sizes. Additionally, conducting evaluations in more scenarios (e.g., mathematical reasoning) to evaluate the impact of AnDPro would be beneficial.

---

> ### Author Rebuttal · Authors · 2025-07-31
>
> Dear Reviewer iGUk,
>
> Thank you for your insightful comments. We sincerely hope that our rebuttal could properly address your concerns.
>
> **Weakness 1. Orthogonality of value vectors.**
>
> > Given that random vectors in high-dimensional spaces are almost orthogonal, and the vector space of value states is a very high-dimensional space, have you observed whether the value state vectors are orthogonal?
>
> Thank you for your suggestion. We highlight our key finding upfront: the value state vectors are not random and nearly orthogonal. Instead, they exhibit correlations. The details are as follows:
>
> - **Value state vectors are not nearly orthogonal---Cosine Similarity Analysis.** We computed the pairwise cosine similarity of 128-dimensional value state vectors extracted from multiple tokens, using the formula:
> $$\text{cos\\_sim}(\mathbf{v}_i, \mathbf{v}_j) = \frac{\mathbf{v}_i \cdot \mathbf{v}_j}{\|\mathbf{v}_i\| \|\mathbf{v}_j\|}.$$
> The results are in the follow table:
>
>    **Table 1:** Pairwise cosine similarity of 128-dimensional value state vectors extracted from five consecutive tokens (T0–T4). Higher similarity is observed between adjacent tokens (e.g., T1–T2, T2–T3).
>
>    | | T0 | T1 | T2 | T3 | T4 |
>    |-|-|-|-|-|-|
>    | T0 | 1 | | | | |
>    | T1 | -0.25 | 1| | | |
>    | T2 | -0.18 | 0.91 | 1  | | |
>    | T3 | -0.22 | 0.72 | 0.91 | 1 | |
>    | T4 | 0.06  | 0.48 | 0.67 | 0.82 | 1 |
>
>    The results show that adjacent tokens exhibit relatively **high** **cosine similarity**, and the overall distribution significantly deviates from zero mean. This indicates that these vectors are not completely random and orthogonal but possess certain correlations and structural properties.
>
> - **Presence of low-dimensional semantic structure---PCA Cumulative Explained Variance** To further assess the structural properties of value vectors, we performed Principal Component Analysis (PCA) on the token value vectors.
>
>   **Table 2:** Explained variance and cumulative variance of the top 10 principal components obtained by performing PCA on 128-dimensional token value vectors.
>    | PC| Explained Variance (%) | Cumulative Variance (%) |
>    |-|-|-|
>    | PC1  | 10.17 | 10.17 |
>    | PC2  | 10.05| 20.22 |
>    | PC3  | 6.46 | 26.67 |
>    | PC4  | 6.37 | 33.04 |
>    | PC5  | 4.19 | 37.23|
>    | PC6  | 4.13 | 41.36|
>    | PC7  | 4.09 | 45.45|
>    | PC8  | 3.89 | 49.35 |
>    | PC9  | 3.89 | 53.24|
>    | PC10 | 2.59 | 55.83|
>
>    The top 10 principal components (**approximately 20% of dimensions**) explain **over 55.8%** of the total variance. This demonstrates that value state vectors **exhibit pronounced low-dimensional linear correlations and encode semantic information**, rather than behaving as purely high-dimensional random noise.
>
> 3. **AnDPro fully exploits these correlations.** Based on these findings, our method uses anchor projection to fully exploit the subtle yet **important correlations** in the value vectors. By integrating this with attention weight information, it achieves more accurate and robust information representation. Detailed theoretical analysis and supporting experimental evidence have been added in **Appendix C.6.**
>
> 4. **Clarification of Figure 1.** In Figure 1, we illustrate the representation of tensors in a high-dimensional space.  Our method exploits the **subtle correlations present in the value vectors**, leveraging anchor projection to enhance the accuracy and robustness of information representation.
>
> **Weakness 2. Updated during forward pass.**
>
> > In the implementation, is the anchor direction updated during each forward pass of the LLM, and what is the associated computational overhead?
>
> Thank you for your question. First, we would like to clarify that our method **by default** performs the anchor direction update **only once during the prefilling stage**, as it is primarily designed to handle long input scenarios efficiently.
>
> However, it can also be optionally applied **during the decoding stage** at specified steps to further maintain efficient memory utilization when necessary.
>
> - **Computational overhead.** As analyzed in **Figure 7 (Appendix C.4)**, we measured the additional time cost introduced by our approach. The measured runtime values are as follows (unit: seconds/milliseconds):
>
>    **Table 3:** Runtime analysis of different methods during prefilling on sequences of length 64K and 128K. Although AnDPro introduces slightly higher update latency compared to SnapKV and Ada-KV, the additional overhead is negligible relative to the total prefilling time.
>
>    | 64K    | Prefilling (s) | KV update (s) | Layer-wise KV update (ms) |
>    |-|-|-|-|
>    | Full| 9.45| - | - |
>    | SnapKV | 9.50 | 0.10 | 3.05 |
>    | Ada-KV  | 9.88 | 0.43 | 13.28 |
>    | AnDPro | 10.04 | 0.58 | 18.13 |
>
>    | 128K   | Prefilling (s) | KV update (s) | Layer-wise KV update (ms) |
>    |-|-|-|-|
>    | Full   | 28.78  | - | - |
>    | SnapKV | 29.00| 0.11| 3.42|
>    | Ada-KV  | 29.47 | 0.59 | 18.31 |
>    | AnDPro | 29.78 | 0.88 | 27.41 |
>
>    The results show that the additional computation introduced by the anchor direction update is **minimal and does not impact the overall decoding efficiency**.
>
> - **Performance and efficiency on decoding.** To further demonstrate the broader applicability of our method, we conducted additional experiments on **long decoding tasks**, particularly in **LLM reasoning** settings. For a better comparison with the baselines, we additionally implemented the decoding versions of AnDPro and SnapKV, while Ada-KV does not have a decoding version. Specifically, we evaluated our approach on the **AIME24** dataset using **DeepSeek-R1-Distill-Qwen-14B**.
>
>    - **Table 4** reports the **reasoning performance (accuracy, %)** under different KV cache budget settings.
>    - **Table 5** reports the **throughput (tokens/s)** and **maximum supported batch size** (OOM threshold) during 10K-token generation.
>
>    **Table 4.**  Performance (accuracy, %) of different methods on the AIME24 dataset under two KV cache budget settings (2K and 4K). The performance of the **full cache** method is **66.66**. Higher values indicate better reasoning performance. For a budget of 2k, the **KV cache update interval** is 1k, and for a budget of 4k, the **update interval** is 2k.
>
>    | **Method** | **Budget = 2K** | **Budget = 4K** |
>    | ---------- | --------------- | --------------- |
>    | SnapKV     | 50.00           | 53.33           |
>    | AnDPro     | **60.00**       | **63.33**       |
>
>    **Table 5.** Throughput (tokens/s) and maximum supported batch size (before out-of-memory occurs) for different methods during 10K-token generation on a single GPU with 80GB of VRAM. Larger batch size and higher throughput indicate better decoding efficiency.
>
>    | Method | Generation Length | Max Batch Size (OOM Threshold) | Throughput (tokens/s) |
>    | - | -| -| -|
>    | Full | 10K | 28 | 181.09 |
>    | SnapKV     | 10K | 80 | 415.48 |
>    | AnDPro     | 10K | 80 | 413.15 |
>
>    The results show that **AnDPro maintains strong reasoning performance under constrained budgets** and achieves **decoding efficiency comparable to SnapKV**.
>
> **Weakness 3. Anchor direction.**
>
> > Could the compressed KV cache mislead the anchor direction compared to its uncompressed state?
>
> - **Compute once**: During the **prefilling stage**, we first compute the anchor direction using the **uncompressed** value states and attention weights.
>
> - **Retain largest projections**: Tokens with the largest projections along this anchor direction are then selected, and only these tokens’ KV entries are retained.
>
> - **Fixed anchor direction**: After this one-time compression, the **anchor direction remains fixed** and is not updated during the decoding stage. The subsequent decoding process uses the compressed KV cache solely for efficient inference.
>
> Because the anchor direction is derived from the **full, uncompressed KV cache**, the compression step does not introduce any bias or deviation from the theoretical derivation presented in Section 3.
>
> **Weakness 4. Experiments on more models with different sizes and more scenarios.**
>
> > Experiments were only conducted on Mistral-7B-Instruct-v0.2, which is insufficient to demonstrate that the method can generalize to other series of models or models of different sizes. Additionally, conducting evaluations in more scenarios (e.g., mathematical reasoning) to evaluate the impact of AnDPro would be beneficial.
>
> Thank you for your suggestions. We have supplemented our experiments with additional results based on the LLaMA and Qwen model families, which are among the current mainstream architectures. The results demonstrate that AnDPro generalizes well across different model architectures.
>
> Furthermore, to comprehensively address your concerns, we conducted two additional sets of experiments:
>
> - **Different model scales.** We extended the evaluation to larger models by testing on Qwen2.5-32B-Instruct using the LongBench single-document QA benchmark. The results show that AnDPro maintains excellent performance at larger model scales.
>
>    **Table 6:** Performance on LongBench single-document QA benchmark using Qwen2.5-32B-Instruct. Results show that AnDPro consistently outperforms Ada-KV across different budget sizes.
>
>    | Method| Budget=128   | Budget=256   | Budget=512   | Budget=1024  |
>    |-|-|-|-|-|
>    | Ada-KV  | 34.66 | 38.02 | 41.57 | 43.82 |
>    | AnDPro | 38.63 | 40.91 | 43.62 | 45.81 |
>
> - **Reasoning scenarios.** We evaluated performance on reasoning tasks using reasoning models. Considering the longer outputs in such tasks, we optimized our implementation to perform online updates during decoding with a budget of 2K tokens and a step size of 1K tokens. The experiments in **Table 4** indicate that AnDPro also exhibits significant advantages in long-form reasoning scenarios such as mathematical reasoning.
>
> We sincerely appreciate your thoughtful and insightful comments and kindly ask you to consider these updates during your further review.

---

> > ### Comment · Reviewer_iGUk · 2025-08-05
> >
> > Thank you to the authors for the detailed and comprehensive rebuttal.
> >
> > The additional analyses on the non-orthogonality of value vectors, the clarification on the one-time calculation of the anchor direction during pre-filling, and the breakdown of the computational overhead have effectively addressed my primary concerns. I especially appreciate the new experiments conducted on the LLaMA and Qwen models, including a larger 32B model and an evaluation on a reasoning task.
> >
> > The work is of high quality, and the rebuttal has substantially improved its clarity and scope. I will raise the quality and clarity scores and remain the overall recommendation.

---

> > > ### Author Response · Authors · 2025-08-05
> > >
> > > Dear Reviewer iGUk,
> > >
> > > Thanks for your kind support and for helping us improve the paper. We sincerely appreciate your valuable suggestions.
> > >
> > > Best regards,
> > >
> > > Authors

---

### Official Review · Reviewer_Z3QZ · 2025-07-03

**Clarity:** 2
**Significance:** 3
**Originality:** 3
**Rating:** 4
**Confidence:** 3

**Summary:**

This paper aims to design an improved scoring policy for KV cache eviction that leverages error in the output to identify the optimal KV cache set to retain. Their approach is complementary to existing KV cache compression strategies. They first present theoretical analysis for the assignment problem, demonstrating how an ideal method would select the optimal set to minimize the degradation in the output vector. They then propose a computationally efficient method that projects value vectors weighted by attention score onto anchor direction (the original unperturbed output) to better estimate token importance. Their approach leads to a reduction in accuracy degradation relative to existing KV cache approaches, without adding substantial noticeable inference efficiency overheads.

**Questions:**

- Is there any guarantee on how close the solution obtained through projection is to the optimal solution from the initial problem (P0-C or P1)?

**Ethical Concerns:**

["NO or VERY MINOR ethics concerns only"]

**Final Justification:**

The author's response has addressed my main concerns around long generation tasks and the benefits of their chunking method. I will maintain my score of 4 as an overall positive assessment of the work.

**Limitations:**

yes

**Paper Formatting Concerns:**

For Figure 5, the decoding latency is hard to assess due to the large latency of the full KV cache baseline. It would be helpful to add a separate visualization comparing different eviction methods with the baseline KV runtime removed to better visualize the discrepancies between different methods.

**Quality:**

3

**Strengths And Weaknesses:**

Strengths:
- They present a novel strategy based on minimizing output degradation which leverages strong theoretical foundations to identify the optimal key set to retain.
- They present a computationally efficient method for identifying the optimal key set to retain based on projecting the attention score-weighted value vectors onto an anchor direction (the original output vector).
- They outperform a range of strong baseline methods (as evaluated on LongBench + Passkey Retrieval)
- Their approach has low overhead for the Update KV phase, as highlighted in Appendix C.4.

Weaknesses:
- Their method is only applicable to compressing the prefill KV cache for generation. It is unclear whether the update step is fast enough to run online during generation.
- They evaluate their method only on short decoding tasks (e.g. LongBench). Some analysis of performance on longer decoding tasks (e.g. LLM reasoning) would be useful for understanding the broader applicability of their method.
- The KV cache merging method (which is similar to pooling) is not well justified - the ablations in C.5 are hard to follow. Some explanation of this approach (or empirical evaluation) would be helpful to support this, including a comparison between chunking and pooling methods as well as intuitive explanation for why this is helpful and superior to pooling.

---

> ### Author Rebuttal · Authors · 2025-07-31
>
> Dear Reviewer Z3QZ,
>
> Thank you for your insightful comments. We sincerely hope that our rebuttal could properly address your concerns. If so, we would deeply appreciate it if you consider raising your score.
>
> **Weakness 1 & 2. Online Eviction and Long Decoding Tasks.**
>
> > Their method is only applicable to compressing the prefill KV cache for generation. It is unclear whether the update step is fast enough to run online during generation.
> >
> > They evaluate their method only on short decoding tasks (e.g. LongBench). Some analysis of performance on longer decoding tasks (e.g. LLM reasoning) would be useful for understanding the broader applicability of their method.
>
> - Thank you for your valuable suggestions. We have **extended our approach to the decoding phase** and conducted additional experiments in the **reasoning scenario** that involves **long decoding**.  The details are as follows:
>
>   - **Online KV cache update mechanism.** We implemented an update mechanism specifically for the decoding stage, where the KV cache is dynamically compressed during token generation. Specifically, after every predefined number of decoding steps, we re-apply **AnDPro** to ensure that the size of the KV cache remains constant at the predefined budget size.
>
>   - **Long decoding reasoning task.**  Our method was originally designed for retrieval-oriented scenarios. To further demonstrate its broader applicability, we have conducted additional experiments under LLM reasoning settings.
>
> - We conduct experiments on the AIME24 dataset using the **DeepSeek-R1-Distill-Qwen-14B**. We report the  performance in **Table 1**, and the throughput and max batch size in **Table 2**.
>
>   Table 1: Performance (accuracy, %) of different methods on the AIME24 dataset under two KV cache budget settings (2K and 4K). The performance of the **full cache** method is **66.66**. Higher values indicate better reasoning performance. For a budget of 2k, the **KV cache update interval** is 1k, and for a budget of 4k, the **update interval** is 2k.
>
>   | **method** | **budget=2k** | **budget=4k** |
>   | ---------- | ------------- | ------------- |
>   | Snapkv     | 50.00         | 53.33         |
>   | Andpro     | 60.00         | 63.33         |
>
>   Table 2: Throughput (tokens/s) and maximum supported batch size (before out-of-memory occurs) for different methods during 10K-token generation. Larger batch size and higher throughput indicate better decoding efficiency.
>
>   | Method     | Generation Length | Max Batch Size (OOM Threshold) | Throughput (token/s) |
>   | ---------- | ----------------- | ------------------------------ | -------------------- |
>   | Full cache | 10k               | 28                             | 181.09               |
>   | SnapKV     | 10k               | 80                             | 415.48               |
>   | Andpro     | 10k               | 80                             | 413.15               |
>
>   The results indicate that **our method maintains strong performance and efficiency** even in reasoning-intensive long decoding scenarios.
>
> **Weakness 3.** **KV** **cache merging method.**
>
> > The KV cache merging method (which is similar to pooling) is not well justified - the ablations in C.5 are hard to follow. Some explanation of this approach (or empirical evaluation) would be helpful to support this, including a comparison between chunking and pooling methods as well as intuitive explanation for why this is helpful and superior to pooling.
>
> Thank you for your valuable comments. In response, we provide the following clarifications and additional experimental analyses:
>
> - **Pooling** **Method as a Baseline** The pooling method was introduced as a baseline to encourage **similarity in attention scores among adjacent tokens**, thereby achieving a chunking-like effect during token selection. To illustrate the impact of pooling on token continuity, we conducted additional experiments using the SnapKV framework on the LongBench-Qasper dataset. We quantified the improvement in token continuity by measuring the **average continuity length**, computed as the total number of retained tokens (excluding the window size) divided by the number of separated token segments. The results are shown in Table 3.
>
>   Table 3:  Average consecutive length and task performance under different pooling kernel sizes, a larger pooling kernel size **increases the average consecutive length**.
>
>   | Pooling kernel             | 3     | 5     | 7     | 9     | 11    |
>   | -------------------------- | ----- | ----- | ----- | ----- | ----- |
>   | Average consecutive length | 6.17  | 8.99  | 11.33 | 13.29 | 15.04 |
>   | Result                     | 29.04 | 30.17 | 30.63 | 29.95 | 31.03 |
>
> - **Chunking Method with Attention–Value Guidance**
>
>   - Beyond pooling, we designed a **chunking method** based on our theoretical insights:
>   - The **anchor direction** is influenced not only by attention but also by **value vectors**, and considering only attention-based semantic consistency is insufficient.
>   - Directly pooling high-dimensional value vectors may lead to loss of directional information.
>   - Therefore, we propose a **chunk operation** that combines attention with value inner products. Specifically, token vectors within each chunk are aggregated by summation, leveraging the fact that adjacent tokens often exhibit strong semantic consistency in the value space. This design **preserves both the importance of attention and the directional information of value vectors**, while also reducing the computational cost of token selection.
>
> - To further analyze the impact of chunking on token continuity, we performed experiments under the AnDPro framework, where attention was already processed with the pooling method. The results are presented in Table 4:
>
>   Table 4:  Average consecutive length and task performance under different Chunk size, a larger chunk size **increases the average consecutive length**.
>
>   | Chunk size                 | 1     | 2     | 4     | 8     | 16    |
>   | -------------------------- | ----- | ----- | ----- | ----- | ----- |
>   | Average consecutive length | 5.99  | 10.25 | 15.39 | 23.37 | 36.13 |
>   | Result                     | 31.72 | 32.69 | 32.44 | 31.93 | 30.93 |
>
> **Question 1. Theoretical guarantee.**
>
> > Is there any guarantee on how close the solution obtained through projection is to the optimal solution from the initial problem (P0-C or P1)?
>
> Thank you for your thoughtful question regarding the theoretical guarantee. Due to the **NP-hardness** of the combinatorial problem, we adopt a **heuristic projection-based scoring function**. While we do not provide a formal error bound at this stage, we **formulate KV cache eviction as an optimization problem**, grounded in convex analysis, which enables principled theoretical reasoning and future extensions. We will **investigate tighter theoretical guarantees** in future work.
>
> **Question 2. Presentation of Figure 5.**
>
> > For Figure 5, the decoding latency is hard to assess due to the large latency of the full KV cache baseline. It would be helpful to add a separate visualization comparing different eviction methods with the baseline KV runtime removed to better visualize the discrepancies between different methods.
>
> Thank you for your valuable suggestion. We address your concerns as follows:
>
> - **Improved visualization.** In the revised version, we will add an additional visualization where the runtime of the full KV cache baseline is removed. This will allow for a clearer comparison among different eviction methods without being overshadowed by the large latency of the full KV baseline.
>
> - **Additional experiments.** We conducted further experiments using the **Mistral-7B-Instruct-v0.2** model on the **Needle-in-a-Haystack (NIAH)** benchmark. We measure the **decoding latency** (in seconds) was measured for an output length of 1K under various input lengths (8K–256K tokens) with a budget size of 128. The results are as follows:
>
>   Table5:  AnDPro maintains **comparable runtime efficiency** to AdaKV and remains close to SnapKV across all input lengths.
>
>   | input length | 8K    | 16K   | 32K   | 64K   | 128K  | 256K  |
>   | ------------ | ----- | ----- | ----- | ----- | ----- | ----- |
>   | SnapKV       | 35.10 | 34.72 | 37.15 | 40.49 | 39.40 | 53.69 |
>   | AdaKV        | 39.22 | 38.99 | 39.64 | 43.39 | 44.23 | 55.30 |
>   | AnDPro       | 39.23 | 38.67 | 40.44 | 42.98 | 44.20 | 55.51 |
>
> - **Efficiency analysis.** Despite incorporating a projection-based scoring function, **AnDPro exhibits runtime comparable to AdaKV** and only slightly slower than SnapKV. These results demonstrate that the additional computations introduced by AnDPro do not significantly affect overall decoding efficiency.
>
> - **Future optimizations.** We plan to introduce **engineering optimizations**, including:
>
>   - Parallelizing scoring computations with the prefilling operations, and
>
>   - Implementing optimized **CUDA** **kernels** for cross-head budget allocation.
>
>   These optimizations are expected to further reduce, or even eliminate, the minor additional overhead observed.

---

> ### Comment · Reviewer_Z3QZ · 2025-08-03
>
> I appreciate the author's response. They have addressed my concerns around handling long generation tasks by extending their method, as well as around the benefits of the chunking method over existing pooling approaches.

---

> > ### Author Response · Authors · 2025-08-04
> > **Thank you for your kind support and insightful comments.**
> >
> > Dear Reviewer Z3QZ,
> >
> > Thank you very much for your kind support and insightful comments. We would deeply appreciate it if you could consider raising your score based on our rebuttal. If you have any remaining concerns, we are more than happy to further improve this work.
> >
> > Best regards,
> >
> > Authors

---

### Note · Authors · 2025-08-12

Dear Area Chair and Reviewers,

We are the authors of Paper 27280 and sincerely appreciate your time and effort during the review process. Your constructive feedback has greatly strengthened our contribution to efficient KV cache eviction for large language model inference, and we are grateful that all reviewers recommended acceptance.

Below is a summary of the reviews and our responses:

- **Reviewer Z3QZ** praised our **“novel strategy based on minimizing output degradation”** and highlighted that we **“outperform a range of strong baseline methods”**. To address concerns about online eviction, long decoding tasks, and and related issues, we conducted experiments on long decoding tasks, and provided clarifications with additional experiments on chunking vs. pooling.

- **Reviewer iGUk** praised the **“innovative”** projection-based scoring function and its **“excellent originality”**, along with our **“thorough theoretical analysis.”** In response to concerns about value vector orthogonality and related issues, we examined value vector correlations, clarified anchor direction computation, and expanded evaluations to larger models and reasoning tasks.

- **Reviewer PCLe**  appreciated our **“mathematically grounded approach”** and **“consistent improvement over prior methods”** across diverse tasks and memory budgets. To address concerns about the theoretical gap between P0-C and P1 and other issues, we explained the relaxation strategy, and expanded evaluations.

- **Reviewer qyz2** noted the **“novel and solid theoretical analysis,”** and remarked that the paper is **“easy to follow”**. To address concerns about small-sized LLM evaluations and computational overhead, we conducted experiments on larger models and provided detailed runtime comparisons.

- **Reviewer AuWV** recognized the **“novel projection-based token importance score”** and the **“comprehensive theoretical analysis/proof paired with intuitive explanations”**. To address concerns about implementation details, we clarified anchor direction computation, attention head token selection, token eviction strategy, and chunk size specifications.

We will incorporate these enhancements, along with your recommendations, into the final manuscript. Thank you and all reviewers for your invaluable feedback and expertise, which have greatly helped improve our work. We look forward to your final assessments and are happy to address any further questions.

Best regards,

 Authors

---

### Decision · Program_Chairs · 2025-09-17

**Decision:**

Accept (poster)

**Comment:**

This paper introduces AnDPro, a projection-based scoring function for KV cache eviction in LLMs. The method is novel and mathematically grounded, going beyond attention-weight heuristics by leveraging value vector geometry. Extensive experiments on LongBench, Needle-in-a-Haystack, and reasoning tasks demonstrate strong and consistent improvements over SnapKV, Ada-KV, and other baselines, with minimal overhead. The authors also provided thorough rebuttals with larger-scale evaluations, runtime analyses, and detailed clarifications on orthogonality, anchor computation, and chunking. While some theoretical gaps remain (e.g., relaxation analysis), the contribution is technically solid and impactful. I recommend acceptance.